# A Tale of Two Continents (and a Few Islands): Ecology and Distribution of Late Pleistocene Sloths

**H. Gregory McDonald**

3309 Snowbrush Court, Fort Collins, CO 80521, USA; hgmcdonald@msn.com

**Abstract:** Late Pleistocene sloths were widely distributed and present in a diversity of habitats in South, Central, and North America and some Caribbean Islands. Late Pleistocene sloths include 27 genera in four families Megatheriidae, Megalonychidae, Mylodontidae, and Nothrotheriidae. There is no consensus on the number of valid species. Some sloths have wide geographic distributions and are present on multiple continents while others have a much smaller distribution. Our knowledge of the paleoecology and natural history of the different sloths varies greatly depending on their relative abundance. The wide distribution of sloths and adaptations to different habitats results in several "sloth" faunas with different taxonomic compositions. These generalized faunas can be distinguished geographically as Temperate North America (five genera), Southern Mexico and Central America (five genera), Northern South America (two genera), West Coast of South America (four genera), the Andes and Altiplano (four genera), Brazilian Intertropical Region (nine genera), Pampas-Patagonia and the Caribbean Islands (Cuba, Hispaniola, and Puerto Rico, four genera). Some genera may occur in multiple regions but are represented by different species. These regions also have differences in other mammalian taxa, so the sloths are often in ecological competition with different megaherbivores or preyed on by different carnivores.

**Keywords:** Xenarthra; sloth; Pleistocene; paleoecology; biogeography





## 1. Introduction

During the Neogene the South American mammalian fauna contained a diversity of herbivores, among which were members of the Xenarthra (sloths anteaters and armadillos) of which sloths, glyptodonts, and pampatheres were primarily herbivorous. Sloths became more prominent in faunas by the early Neogene and in the Miocene (Santacrucian SALMA) consisting of a greater taxonomic diversity and a greater range of sizes [1]. While not all sloths were giants or megaherbivores, there is a trend that continued into the late Pleistocene for an increase in size, culminating in multi-ton taxa such as the megatheriids, *Eremotherium* and *Megatherium* and a mylodontid, *Lestodon*. Xenarthrans were usually the largest herbivores in a fauna until the appearance of proboscideans from North America during the Great American Biotic Interchange (GABI).

Of all the taxonomic groups originating in South America, the sloths were the best dispersers, reaching the Caribbean Islands in the middle Neogene prior to the formation of the Isthmus of Panama. All sloth taxa found on Caribbean Islands are members of the Megalonychidae and first appear in the early Miocene in Cuba [2], the late Miocene–early Pliocene in the Dominican Republic, on Hispaniola [3], and early Oligocene of Puerto Rico [2]. By the late Miocene, a megalonychid and a mylodontid are present in North America, again prior to the establishment of the Isthmus of Panama, and this was followed by later separate dispersals of sloths including nothrotheriid, megatheriids as well as additional megalonychids and mylodontids [4]. As a result of these dispersal events sloths became established in faunas in Central America, North America, and some Caribbean islands until their extinction there as well as in South America. While an exact chronology

of the extinction of individual sloth taxa is not known, radiocarbon dating of the Caribbean taxa shows they survived later than their continental counterparts [5].

The presence of sloths in different geographic areas resulted in the creation of varied sloth communities as a direct result of the founder species, additional dispersals at different times, local extinctions, and evolutionary diversification in the different regions (Table 1). Some of the taxa are restricted to small areas and may be considered endemics, while other taxa had wide ranges which may include South, Central, and North America. The taxonomic composition of these sloth communities may vary at the genus or species level. The presence of multiple sloth taxa in a local fauna reflects the wide range of differences in the ecology of each species and the adaptation to different niches and habitats. This diversification is also reflected in the range of sizes of sloth taxa, and despite the popular sobriquet of the giant to describe the extinct sloths, this term is not applicable to all fossil sloths, although many of them do fall into the category of being a megaherbivore with body masses exceeding 100 kg. Diversification not only allowed sloths to minimize competition with other sloths but also with the herbivores with which they evolved in South America as well as the native herbivores in the regions into which they dispersed.

**Table 1.** Distribution of sloths in different geographical areas. Temperate North America (TNA), Southern Mexico and Central America (SMCA), Northern South America (NSA), West Coast of South America (WCSA), Andes and Altiplano (AA), Brazilian Intertropical Region (BIR), Pampas-Patagonia (PP), Caribbean Islands (CI). C—Cuba, H—Hispaniola, P—Puerto Rico. G—genus present but species not known.

| TAXON | TNA | SMCA | NSA | WCSA | AA | BIR | PP | CI |
|---|---|---|---|---|---|---|---|---|
| Megatheriidae | | | | | | | | |
| *Megatherium americanum* | | | | | | | X | |
| *Megatherium celendinense* | | | | | X | | | |
| *Megatherium medinae* | | | | | X | | | |
| *Megatherium sundti* | | | | | X | | | |
| *Megatherium tarijense* | | | | | X | | | |
| *Megatherium urbinai* | | | | | X | | | |
| *Eremotherium laurillardi* | X | X | X | X | | X | | |
| Megalonychidae | | | | | | | | |
| *Megalonyx jeffersonii* | X | | | | | | | |
| *Nohochichak xibalbahkah* | | X | | | | | | |
| *Xibalbaonyx oviceps* | | X | G | | | | | |
| *Xibalbaonyx exinferis* | | X | | | | | | |
| *Xibalbaonyx microcaninus* | X | | | | | | | |
| *Meizonyx salvadorensis* | | X | | | | | | |
| *Megistonyx oreobios* | | | | | X | | | |
| *Diabolotherium nordenskioldi* | | | | | X | | | |
| *Ahytherium aureum* | | | | | | X | | |
| *Australonyx aquae* | | | | | | X | | |
| *Megalocnus rodens* | | | | | | | | X (H) |
| *Parocnus serus* | | | | | | | | X (H) |
| *Parocnus browni* | | | | | | | | X (C) |
| *Parocnus dominicanus* | | | | | | | | X (H) |
| *Acratocnus odontrigonus* | | | | | | | | X (P) |
| *Acratocnus ye* | | | | | | | | X (H) |
| *Acratocnus simorhynchus* | | | | | | | | X (H) |

**Table 1.** *Cont.*

| TAXON | TNA | SMCA | NSA | WCSA | AA | BIR | PP | CI |
|---|---|---|---|---|---|---|---|---|
| *Acratocnus antillensis* | | | | | | | | X (C) |
| *Neocnus comes* | | | | | | | | X (H) |
| *Neocnus dousman* | | | | | | | | X (H) |
| *Neocnus toupiti* | | | | | | | | X (H) |
| *Neocnus major* | | | | | | | | X (C) |
| *Neocnus gliriformis* | | | | | | | | X (C) |
| Mylodontidae | | | | | | | | |
| *Glossotherium robustum* | | | | | | | X | |
| *Glossotherium tropicorum* | | | | X | | | | |
| *Glossotherium phoenesis* | | | | | | X | | |
| *Mylodon darwinii* | | | | | | | X | |
| *Lestodon armatus* | | | | | | | X | |
| *Paramylodon harlani* | X | | | | | | | |
| *Oreomylodon wegneri* | | | | | X | | | |
| *Ocnotherium giganteum* | | | | | | X | | |
| *Mylodonopsis ibseni* | | | | | | X | | |
| *Scelidotherium leptocephalum* | | | | | | | X | |
| *Catonyx cuvieri* | | | | | | X | | |
| *Catonyx chiliense* | | | | X | | | | |
| *Valgipes bucklandi* | | | | | | X | | |
| Nothrotheriidae | | | | | | | | |
| *Nothrotheriops shastensis* | X | X | | | | | G | |
| *Nothropus priscus* | | | | | | | X | |
| *Nothrotherium maquinense* | | | | | | X | X | |

For purposes of comparison of the sloth diversity in different regions, the following regions with good documentation of the sloth fauna are used: Temperate North America, Southern Mexico and Central America, Northern South America, Coastal South America (west of the Andes), the Andes and Altiplano, Brazilian Intertropical Region, Pampas-Patagonia, and the Caribbean Islands (specifically Cuba, Hispaniola, and Puerto Rico). These regions only correspond to areas with a distinctive sloth fauna and not to the Last Glacial Maximum (LGM) biomes proposed by [6–9] nor to any previously defined physiographic areas except in a general way. In many places, such as Central America, knowledge of the sloth fauna is limited and there are other large areas for which the late Pleistocene fauna is unknown so finer resolution is not possible. Hence while the choice of these broad regions is arbitrary, it is functional in examining the association of different sloth taxa in a region with the full recognition that further refinement will be needed. The one exception is the Brazilian Intertropical Region (BIR) as defined by Cartelle [10] as it is one of the few established biogeographic areas based on its late Pleistocene fauna in South America and has many known faunas that include sloths that have been studied. Gallo et al. [11] in their study of distributional patterns of the Pleistocene megaherbivores of South America identified several generalized tracks and nodes. Their GT3 or transandine corridor coincides with a distribution first proposed by Moreno et al. [12]. While their track G2 which includes the Andes in Peru and Bolivia I extend it north into Ecuador and Venezuela as a single contiguous region. Track G4 of Gallo et al. [11] covers the Pampas to which I have included Patagonia and their tracks GT5 and 6 correspond to the BIR. As the study of Gallo et al. [11] was restricted to South America so there are no corresponding general tracks for Central and North America or the Caribbean.

Varela et al. [13] identified the existence of provincialization within the xenarthran megamammals which grouped into at least three bioregions. Northern and southwestern taxa overlap in the Río de la Plata region where also some endemic taxa are found. The study of potential areas of distribution, as well as that of potential areas of co-occurrence of extinct species with equal or different ecological requirements, represents a valuable tool to infer paleoenvironmental conditions, interactions between species, and their evolutionary history.

## 2. Regions with Distinct Sloth Faunas

### 2.1. Temperate North America

This area covers Canada, the United States, and the area between the Sierra Madre Oriental and Occidental in Mexico north of the Isthmus of Tehuantepec.

### 2.2. Southern Mexico and Central America

This area extends from the Isthmus of Tehuantepec in Mexico south through Central America to the connection of Panama with South America.

### 2.3. Northern South America

The Northern South American region includes Venezuela, Trinidad, Colombia, and Guyana. Although the northern Andes extend into Venezuela and Colombia, sloths from higher elevations are included in the Andean Altiplano region.

### 2.4. West Coast of South America

The region between the western flank of the Andes and the Pacific Ocean extending from the juncture of Panama and Colombia south to Tierra del Fuego, Chile.

### 2.5. Brazilian Intertropical Region

The Brazilian Intertropical Region includes two biogeographical subregions, intertropical and temperate, that are distinguished for the Pleistocene of Brazil combined. The BIR was proposed and defined by Cartelle [10] as a distinct zoogeographical domain, based on the occurrence of endemic species, including Pleistocene taxa, from the Brazilian states of Goiás (GO), Minas Gerais (MG), Rio de Janeiro (RJ), Espírito Santo (ES), Bahia (BA), Sergipe (SE), Alagoas (AL), Pernambuco (PE), Rio Grande do Norte (RN), Paraíba (PB), Ceará (CE), and Piauí (PI). The Pleistocene fauna from this part of Brazil has been studied in more detail than other parts of tropical South America, so serves as a proxy for the region in general. The late Pleistocene fauna includes nine sloth taxa.

The temperate region includes areas extending south from the present Tropic of Capricorn and encompasses the states of Rio Grande do Sul, Santa Catarina, Parana, and the southern part of the state of São Paulo. The composition of the sloth fauna from this area resembles those in Argentina, Paraguay, and Uruguay and differs from the sloth fauna present in intertropical Brazil. Some sloths have wide distributions that may include multiple regions such as *Eremotherium laurillardi*, which is present as far south as Rio Grande do Sul [14,15] and north into Central and North America, and the scelidotheriine, *Catonyx cuvieri*, which has been reported from Uruguay [16] but is more common in predominantly intertropical regions.

### 2.6. Andes and Altiplano

The Andes extend from Venezuela and Colombia (ca. in the North to Southern Chile and form a major biogeographic behavior in Eastern and Western South America. The range is 8900 km long and extends through Venezuela, Colombia, Ecuador, Peru, Bolivia, Chile, and Argentina. The average height is about 4000 m. The Altiplano is formed by the broadening of the Andes above 3000 masl between 13° and 27° S.

## 2.7. Pampas and Patagonia

The Pampas is one of the largest grassland plains in the world, covering almost one-third of Argentina's land area. The Pampas extends from the Atlantic to the Andes and continues into Uruguay and Brazil's southernmost state, Rio Grande do Sul. Patagonia is contiguous with and extends south from the Pampas from 39° S to 55° S.

## 3. Sloth Paleoecology

Despite most extinct sloths being referred to as giants, there is an impressive range in body mass of late Pleistocene sloths (Figure 1). While there have been multiple studies of body mass estimates of sloths [17–20] the resulting estimates of body mass by different workers have varied and have been limited to the more common genera and species, along with some estimates for taxa for which the mass has not been previously estimated.

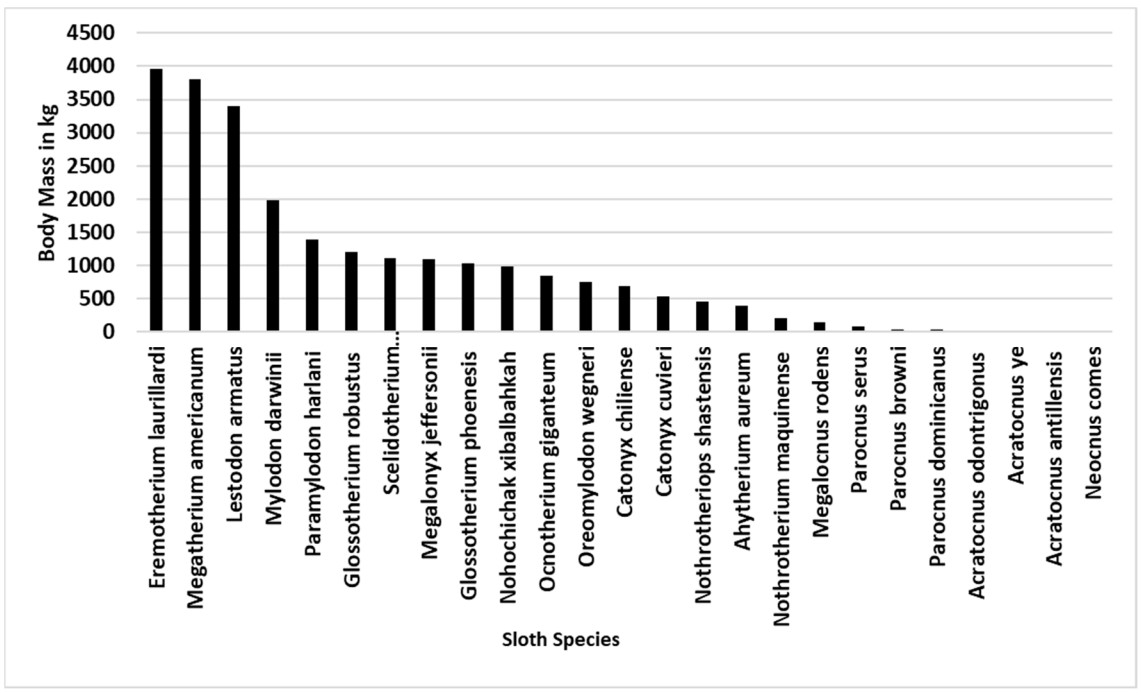

**Figure 1.** Comparison of the estimated body mass of late Pleistocene sloths. As the estimated weight of *Acratocnus antillensis* and *A. ye* is 15 kg, *A. antillensis* is 10 kg and *Neocnus comes* is 8.36 kg they are not visible at this scale.

For many of the sloth taxa included here, there are no previous body mass estimates. Included in the summary of the ecology of the different sloths are preliminary body mass estimates for taxa for which a complete femur is known is included. The estimate is based on the formula of Fariña and Vizcaino [21] on the relationship of femur length to body mass in armadillos that scales closely to that of generalized mammals, while other bones such as the humerus scale differently because of its functional tie to digging. McDonald [19] used this approach to provide a consistent way to compare the body mass of the North American fossil xenarthrans as the choice of a single element reduces potential error generated by utilizing different skeletal elements in different taxa. This information provides a preliminary basis for a comparison of the body mass of the different genera and species and makes the data used in comparisons consistent in identifying trends in size change in the various lineages chronologically or geographically. The choice of this formula, (Formula F1 of Fariña et al. [17] and Scott [22] which uses the length of the femur between proximal and distal articular surfaces to calculate body mass has the added advantage that it generates a body mass estimate from a part of the skeleton that can be compared with those calculated for South American late Pleistocene xenarthran taxa based on other skeletal

elements. The formula utilized is: log10mass = 3.485 × log10 (femur length in cm) – 2.9112. The femora lengths used in calculating body mass provided here are derived from both personal measurements of specimens and from the literature. When possible, the averages of specimens were used, but in some cases, only a single specimen is available for some taxa, particularly taxa in which the holotype is cranial material. While the measurements included here provide an initial approach to compare body mass estimates of the taxa in this review, more refined studies will be needed to provide a more solid basis for body mass estimates for future studies.

Available paleoecological information for the different late Pleistocene sloth taxa varies. For example, dietary interpretation based on carbon and nitrogen isotopic analysis is not known for many taxa while it is well known for other taxa such as *E. laurillardi*, *C. cuvieri*, *V. bucklandi*, and *N. maquinense* so often there is no comparable isotopic data for all species in a region. Inferences of diet can also be made based on morphological characters and indices such as the relative muzzle width and occlusal surface area which have been used to provide a general interpretation of the differences in the paleoecology of these species [23]. This approach is limited as it depends on the completeness of the skulls of the sloth, so it is like isotopic data in that it is not always available for all taxa. In some taxa, the data provided by the relative muzzle width index and the occlusal surface area is available such as *Eremotherium laurillardi*, *Ocnotherium giganteum*, *Mylodonopsis ibseni*, *Glossotherium phoenesis*, *Catonyx cuvieri*, *Valgipes bucklandi*, *Ahytherium aureum*, and *Australonyx aquae* and in the case of *E. laurillardi*, *C. cuvieri* and *V. bucklandi* complements the carbon isotopic data and stereomicrowear analysis indicating they were adapted to a mixed-feeder diet, varying in the consumption of C3 and C4 plants.

## 4. Synoptic Overview of Late Pleistocene Ground Sloths

### 4.1. Family Megatheriidae

#### 4.1.1. *Megatherium americanum*—Pampas (Figure 2A)

The type of *Megatherium americanum*, the first fossil sloth described, came from the bank of the Luján River near the city of the same name located 65 km west of Buenos Aires, Buenos Aires Province, Argentina. Estimates of the body mass of *Megatherium* in the literature vary, with [17] obtaining ranges from 3 to 6 tons, 4000 kg [24,25] proposing 3800 kg based on scale models. This last value is close to the 3706 kg obtained by Brassey and Gardiner [26] using a shape-fitting algorithm based on a mounted skeleton. Multiple species of *Megatherium* have been described and the genus is divided into two subgenera, *Megatherium* (*Megatherium*) with two species, *M. americanum* and *M. altiplanicum* and *M.* (*Pseudomegatherium*) with multiple species associated with the Andes Altiplano region.

The distribution of *M. americanum* is restricted to the eastern side of the Andes and at lower elevations. Most records are from the Pampas of Northern Argentina and south to the northern part of Patagonia, into Southern Bolivia, Uruguay, and Paraguay and extending into the southeastern part of Brazil.

Analysis of the skull of *Megatherium americanum* [27] indicates it was adapted for strong and vertical biting. The teeth are extremely hypsodont and bilophodont, with each loph forming a sharp cutting edge. This morphology suggests the teeth were used for cutting, rather than grinding, and that hard and fibrous food was not the main dietary component. Dung referred to this species is not available so inferences as to the diet of *M. americanum* merits more rigorous analysis. The current morphological data does indicate that it had a browsing diet in open habitats but also could have fed on moderate to soft tough food.

The $\delta^{13}$C values for *M. americanum* from the LGM site, Santa Rosa, indicate it consumed exclusively C3 plants from open areas. The minimum value of $\delta^{13}$C for *M. americanum* is 8.7‰ and $\delta^{13}$C values of *Megatherium* from Playa del Barco, also from the LGM, also indicate *Megatherium* was feeding on intermediate C3–C4 open vegetation [28].

High $\delta^{15}$N values measured on bone collagen from some individuals of *M. americanum*, have been attributed to the consumption of plants with high $\delta^{15}$N values that grow under arid

conditions [29,30]. The $\delta^{13}$Ccarb-coll values of the three specimens of *Megatherium*, examined by Bocherens et al. [31] for which carnivory was suggested [24,32], clearly fall in the same range as the large herbivores such as the equid *Hippidion*, the notoungulate *Toxodon* and the liptoptern *Macrauchenia*, for which there is little doubt they had an herbivorous diet.

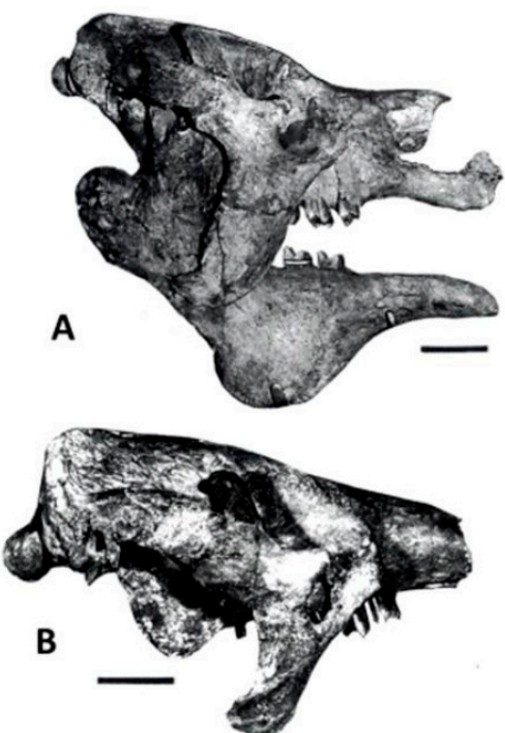

**Figure 2.** Family Megatheriidae. *Megatherium americanum* (**A**) and *Eremotherium laurillardi* (**B**).

The $\delta^{18}$O values for *Megatherium* from La Pampa Province, Argentina showed the lowest value (27.7‰). The mean average temperature (MAT) for the LGM sites of Santa Rosa and Playa del Barco calculated from the $\delta^{18}$O$_{PO4}$ values obtained from *Equus* was 14.8 °C.

The reconstruction of the vegetation and climate of the Pampa grassland in Argentina, during the late Quaternary was based on pollen recovered from dated stratigraphic sections from arroyo walls and archaeological excavations. The palynological evidence from the Pampas suggests that during the late Pleistocene, the vegetation was primarily composed of herbaceous psammophytic species characteristic of steppe environments with associated xerophytic woodland in the southwestern part of the Pampas, and [33] and references therein. This evidence suggests an environment dominated by hard grasses, with less abundant moderate to soft tough plant material, along the distributional range of *M. americanum*. This habitat existed prior to 10,500 years B.P. after which there was a change towards vegetation characteristic of ponds, swamps, and floodplains, or toward environments with locally more effective moisture, suggesting annual precipitation close to modern levels or a higher availability of water in the central part of the Pampa grassland.

### 4.1.2. *Megatherium celendinense*—Andes and Altiplano

The type and only known locality for this species is Santa Rosa cave, located near the city of Celendín (Cajamarca Department, Peru) at 2625 m. The femur is not known for this species so no estimate of body mass is available but based on other parts of the skeleton it is the largest Andean megatheriine equivalent to *M. americanum*, and *E. laurillardi* in size [34]. *M. celendinense* is one of several described species that form an Andean lineage of *Megatherium* (*Pseudomegatherium*) (*M. sundti*, *M. urbinai*, *M. tarijense*, and *M. medinae*) that is distinct from *M.* (*Megatherium*) *altiplanicum* and *M.* (*Megatherium*) *americanum* which inhabited lower elevations.

Along with *Megatherium celendinense* two medium-sized megatheriids from Pleistocene deposits, one from near Ulloma, Bolivia (ca. 3810 m elevation), *Megatherium sundti* and *Megatherium medinae* from Pampa del Tamarugal, near Pica, Tarapacá Province, Chile (ca. 1310 m) have been described [35]. The two species are distinguished primarily based on differences in the femur with *M. medinae* having a more plesiomorphic morphology while in *M. sundti* the femur more closely resembles the derived condition in *Megatherium americanum* [36]). Based on femur length the estimated body mass of *M. sundti* is 1253 kg. There are less pronounced morphological differences in the skull, mandible, and tibia associated with these distinct femoral morphologies. The differences are sufficient to distinguish *M. sundti* as a valid species, distinct from *M. medinae* and another medium-sized megatheriid, *Megatherium tarijense* from Pleistocene deposits near Tarija, Bolivia. Most studies of these species have focused on phylogenetic relationships, and nothing has been proposed relative to their paleoecology. The study by Gallo et al. [12] identified a distributional track for *M. medinae* in the Andes that is distinct from *M. americanum*.

4.1.3. *Megatherium urbinai*—West Coast of South America

The type locality for this species is from the Peruvian coast, near North Sacaco, Arequipa Department, Peru, elevation of 100 m and it is also reported from Tres Ventanas Cave, east-south-east of Lima, at an elevation of 4000 m. It is a small to medium-sized *Megatherium* species closely resembling the other Andean Megatheriinae (*M. medinae*, *M. tarijense*, and *M. elenense*) [37].

*Eremotherium laurillardi*—Temperate North America, Southern Mexico, and Central America, Northern South America, West Coast of South America, Brazilian Intertropical Region (Figure 2B).

*Eremotherium laurillardi* has been referred to as the Pan–American ground sloth as it has the widest latitudinal distribution of all sloths [38]. The late Pleistocene range of extens from the state of Rio Grande do Sul (30.5° S) in Southern Brazil to New Jersey (ca. 40.3° N) in the United States. The northern extension of the range into New Jersey in the United States reflects a temporary northern expansion of the species range during the last interglacial. Most Rancholabrean records of *Eremotherium* in the United States are from the Sangamonian interglacial with one record from the earliest Wisconsinan [39]. It is possible that the range of *Eremotherium* in the United States had contracted southward by the earliest Wisconsinan and the species may not have been present in the United States by the late Pleistocene although it still had an extensive range from Mexico to Southern Brazil at the time of its extinction. *Eremotherium* is the most common sloth in the Southern Mexico and Central America region, reflecting the large size of its bones and making it more noticeable in the field. Its distribution includes one record from Belize, eight from Guatemala, three from Costa Rica, three from El Salvador, one from Nicaragua, four from Honduras, and three from Panama.

*Eremotherium laurillardi* has the highest estimated body mass, 3961 kg, of any species of extinct sloth.

The isotopic data for *Eremotherium* and other associated mammalian taxa indicates a dry environment with mixed vegetation with a predominance of open savanna, with shrubs and low–density forests suggesting that the area at the time was like the current habitat of arid and open areas of shrub savanna. Available oxygen isotopic data from speleothems in Northeastern Brazil [40] show a decrease in humidity between 27 ka to 21 ka, and an increase in humidity between 21 ka to 10 ka, being, on average, a dry interval in the late Pleistocene. This is complemented by oxygen isotopic data associated with radiocarbon dates from 27–10 ka, which although punctuated also overall show homogeneous dry climatic conditions in this region during the LGM.

Stable isotope analysis of an *Eremotherium* tooth from Belize from the Last Glacial Maximum (LGM) (26,975 ± 120 calendric years) along its entire length for carbon and oxygen showed the tooth recorded two wet seasons separated by one long dry season and that this sloth was able to adapt its diet to survive the marked seasonality of the LGM. The higher

average $\delta^{13}$C value of $-6.8$‰ (intermediate between $-9$ and $-2$‰) of the specimen indicates more of a mixed diet and indicates the species was both browsing and grazing [41]. *Eremotherium* in this region relied more on C4 or CAM vegetation during the wet season, and C3 plants during the dry season, consistent with its hypothesized adaptive flexibility.

In Northern South America *Eremotherium* is known from Venezuela (seven localities), Trinidad (one locality), Colombia (three localities) [42], and Guyana (two localities). Depositional environments with *Eremotherium* in Venezuela and Trinidad include tar deposits [43,44].

The remains of at least 22 individuals of *Eremotherium laurillardi* were recovered from the Tanque Loma locality on the southwest coast of Ecuador. Multiple lines of evidence suggest that these sloths may have congregated and died in a mass mortality event in a marshy riparian habitat. The sediments suggest the original environment of deposition was a low-energy anoxic aquatic environment and contained abundant plant material consistent with digested fodder representing coprolites or gut contents of *E. laurillardi* were secondarily infiltrated by asphalt [45]. Other sites with *E. laurillardi* from the region include Machalilla further north on the Ecuadorian coast [46], Rio Engabao [47], and Cautivo [48] have yielded isolated elements pertaining to *Eremotherium*.

Most Brazilian records of *Eremotherium laurillardi* are from the Brazilian Intertropical Region with 62 localities. It is widely distributed throughout the north (Acre and Rondônia), northeast (all states), midwest (Goiás, Mato Grosso, and Mato Grosso do Sul), and southeast (all states) of Brazil but becomes less common in the south.

In the BIR stable isotope analysis of *E. laurillardi* from Sergipe State, Northeastern Brazil between 11,084 and 27,690 calendric years, indicated that as in other parts of the species range it was also a mixed feeder diet utilizing both C3/C4 plants based on $\delta^{13}$C values of –7.7 to –3.3 [49]. These values indicate the presence of C3 plants, such as herbs and shrubs, so the meso–megamammals from the late Pleistocene of Sergipe lived in a more closed and drier environment than previously occurred in Sergipe between 27 to 11 ka. Comparable results based on 40 samples for *Eremotherium* from the BIR [50] included 24 new isotopic data which were incorporated with data previously published [51–53]. The $\delta^{13}$C range was between $-11.01$‰ and $0.41$‰, with a mean value of $-5.66 \pm 2.60$‰. The results indicated that *Eremotherium* was a generalist species, but in this region, its diet was dominated by C4 plants (pi = 53%) rather than C3 plants (pi = 47%). The $\delta^{18}$O varied between $27.62$‰ and $32.77$‰, [51–53]. The diet of *Eremotherium*, as a mixed feeder, did not change between 27 and 10 ka, based on associated radiocarbon dates. The inferred diet for *E. laurillardi* based on relative maxillary width is consistent with the carbon isotopic data [50] and stereomicrowear analysis [52], suggesting a mixed-feeder diet, with a little higher consumption of C3 plants (piC3 = 52‰).

### 4.2. Family Megalonychidae

#### 4.2.1. *Megalonyx jeffersonii*—Temperate North America (Figure 3A)

After *Eremotherium*, *Megalonyx* has the second greatest latitudinal range of the North American late Pleistocene ground sloths extending from the Yukon [53] South to Central Mexico. Changes in its distribution during the Pleistocene reflect its response to changes in climate with the more northern occurrences and presence at higher elevations occurring during the Sangamonian interglacial while during the Wisconsinan the range contracted to lower latitudes and elevations.

The estimated mean body mass of *M. jeffersonii* in the late Pleistocene is 1090 kg although some individuals were markedly larger such as an inferred female from a Sangamonian interglacial site in Iowa with an estimated body mass of 1286 kg [54].

Stable isotope analysis of *Megalonyx* from New York showed a $\delta^{13}$C value of $-20.5$‰ and implied a diet of only C3 plants. The $\delta^{15}$N value was 5.0‰ so is like that of modern herbivores of New York [55]. These isotopic values are also comparable to values previously obtained for this species from the late Pleistocene Saltville locality, Virginia, supporting the idea that this species was a forest-dwelling browser [56].

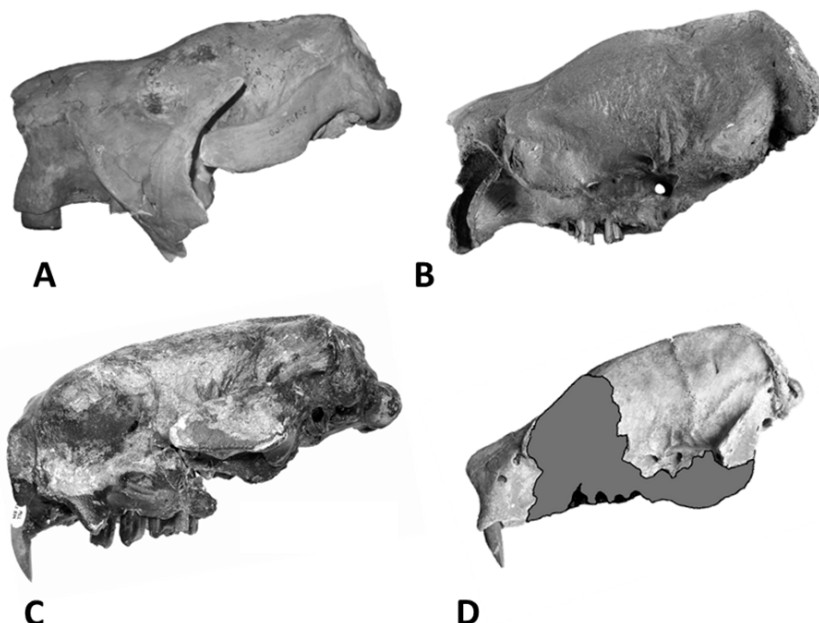

**Figure 3.** Family Megalonychidae. Continental taxa. *Megalonyx jeffersonii* (**A**), *Megistonyx oreobios* (**B**), *Ahytherium aureum* (**C**), *Australonyx aquae* (**D**).

### 4.2.2. *Nohochichak xibalbahkah*—Southern Mexico and Central America

This monotypic genus is currently known only from a single locality, the Hoyo Negro portion of the Sac Actun cave system in the Eastern Yucatán Peninsula, Quintana Roo, Mexico. The cave entrance is at 9 masl. During the LGM, the lowering of the sea level and the drop in the local water table in the area around the cave system would have resulted in a more arid landscape. The current vegetation on the Yucatán is profoundly affected by the alternating dry and wet seasons, and vegetation zones gradually grade into one another horizontally and lack sharp boundaries due to the lack of any mountain ranges. Tropical dry forests extend across the north–central portion while the distribution of tropical forests composed of deciduous semi-evergreen forests is present in the central, eastern, and southern parts of the Yucatán Peninsula [57]. The pattern of distribution of the vegetation reflects the pattern of rain distribution which ranges from the driest areas in the northeast to the most humid areas in the southwest [58]. Located in the southeastern portion of the Yucatán Peninsula, the vegetation around Actun Sac today is tropical forest.

*Nohochichak* is only known from the late Pleistocene of the Yucatán Peninsula [59]. It is about the size of a medium-sized *Megalonyx* and based on femur length the estimate of its body mass is 987 kg.

### 4.2.3. *Xibalaonyx* (*X. oviceps*, *X. microcaninus*, *X. exinferis*) Temperate North America, Southern Mexico, and Central America

While three species have been described for this genus, each is known only from the holotype, so information on the palaeoecology for the genus is more limited compared to other sloth genera. The holotype of *X. oviceps* is an immature animal as indicated by the unfused epiphyses of all the major limb bones and the visible sutures in the skull along with the general lack of rugosity on the bones commonly present in adult sloths [60]. The femur of the holotype is preserved and retains the unfused head, but the distal epiphysis is absent. A very rough estimate of the body mass of this juvenile is 200 kg which given the ontogenetic immaturity of the skeleton would suggest the adult may have been the largest of the North American Pleistocene megalonychids.

There are no radiometric dates for any of the species, but all are presumed to be late Pleistocene in age. The types of *Xibalbaonyx oviceps* and *X. exinferis* were recovered from adjacent cenotes on the Yucatán Peninsula [61] while *X. microcaninus* is known from the

state of Jalisco. While all three species are from localities that are about the same latitude, there is a significant difference in the elevation of the Jalisco specimen at 1372 m.

### 4.2.4. Aff. *Xibalbaonyx*—Northern South America

Despite the fragmentary nature of the megalonychid from Cueva de Iglesitas near Caracas and the limited number of characters that could be scored, this specimen nested with *Xibalbaonyx* which is otherwise only known from Mexico. During the Last Glacial Maximum, this part of Venezuela was much drier with colder weather, which concentrated humidity at high elevations and at the top of high mountains in the Neotropics [62,63]. The same conditions existed on the Cordillera de la Costa range close to the Río Guaire valley, which is at approximately 1000 masl, so would have also experienced drier and colder conditions, just below the periglacial vegetation zone (1200 masl) reported during the late Pleistocene by Salgado–Laboreau (1979) [64] for the Merida Mountain range, which was probably dominated by a dry to woodland vegetation. As Morro de Iglesitas is a karst region associated with the Río Guaire, during the LGM it may have been a forest refuge, as the limestone may have retained the water originating from more tropical vegetation covering its surface. The probable scenario in the Caracas valley during the late Pleistocene was a mosaic of vegetation types dominated by high dry savannas, with patches of deciduous tropical forest located in the karstic zone. Such a landscape is consistent with the idea that *Xibalbaonyx* in Mexico was flexible in its habitat preferences and was living in areas with both a drier as well as more tropical rain forest.

Remains of other megalonychids are known from three other sites in Venezuela, Cueva del Miedo (Falcón State) based on teeth and part of a femur; Cueva los Escorpiones (Anzoategui State) with teeth and carpals and Cueva Cerro la Chapa (Falcón State), with a fragment of the humerus. An indeterminate megalonychid is also known as a tar deposit, El Breal de Orocual [43]. The limited samples have not permitted identification beyond family, and further study is needed to determine if they can be assigned to aff. *Xibalbaonyx*, *Megistonyx,* or another taxon within the family.

### 4.2.5. *Meizonyx salvadorensis*—Southern Mexico and Central America (Figure 3B)

*Meizonyx salvadorensis* is known from the middle Pleistocene of El Salvador and the late Pleistocene of Mexico. The femur, upon which other body mass estimates were made for the other sloths is not known for this sloth so while it is a large taxon based on a comparison of skull measurements to other sloths [65], it is similar in size to *Megalonyx jeffersonii*.

The distribution of *Meizonyx* in Mexico is limited to two nearby sites in the Sierra Mazateca of the Sierra Madre Oriental del Sur, State of Oaxaca [65]. The height of the Sierra Mazateca ranges from 500–2800 m, resulting in a high diversity of habitats and the two sites are at 1466 and 1599 masl. During the Pleistocene, lower temperatures during glacial intervals resulted in a downslope migration of montane vegetation zones [66–68]. These shifts occurred as cycles of expansion and contraction of mountain pinyon, juniper, and oak woodlands in the mountains and shifts in their elevational distribution. The $\delta^{13}$C value of –23‰ in *Meizonyx* indicates it was feeding on C3 plants in a mesic environment so it may have been living in an oak–pine or oak-dominated habitat which had shifted downslope. Outside of Mexico the only other record of *Meizonyx* is from a lower elevation, 687 masl, in El Salvador [69]. This site is considered middle Pleistocene in age and given the paucity of palaeobotanical data from the site it is difficult to integrate with the Mexican record.

### 4.2.6. *Megistonyx oreobios*—Andes and Altiplano

*Megistonyx* was recovered from a cave near Cerro Pintado, located on the border of Venezuela and Colombia. This mountain is in the Sierra de Perijá Mountain Range, a branch of the Northern Andes, in Zulia State, Venezuela, so is included in the Andean region. The cave is at an elevation of 3200 m. Currently, this sloth is only known from Venezuela and the northern part of South America [70].

The Mérida Glaciation refers to the late Pleistocene glacial episode in the Venezuelan Andes. This glacial event in the northern part of South America corresponds to the Würm-Wisconsin Glacial of the northern hemisphere. The LGM in the region was between ~22,750 and 19,960 calendric years and the paleo-glacier equilibrium-line altitudes in the nearby Cordillera de Mérida are estimated to have been ~850 to 1420 m lower than present. The estimated local LGM temperatures would have been at least 8.8 ± 2 °C cooler than today [71]. The elevation of the cave where the remains of *Megistonyx* were recovered, 3200 m, suggests that the animal was living under at least seasonally cold climatic conditions. The radiocarbon age of *Megistonyx* from Cerro Pintado is 14,150 ± 50 RC years BP (17,383 ± 239 calendric years) which indicates the animal lived at this high elevation after the LGM.

The femur was not recovered so there is no estimate of the body mass, but the animal is comparable in size to *Megalonyx*. There is no stable isotope data for the species.

### 4.2.7. *Diabolotherium nordenskioldi*—Andes and Altiplano

The type locality of *Diabolotherium nordenskioldi* is Casa del Diablo, a cave at an altitude of >3800 masl located on the Southern Peruvian altiplano northwest of Lake Titicaca, Puno Department, Peru. It is also known from the coastal site of Piedra Escrita, Peru, and three other caves in the Peruvian Andes, Jatun Uchco (2150 masl) and Trigo Jirka (2700 masl) in the Departamento de Huánuco and Cueva Roselló (3875 masl), Departamento de Junín [72]. Outside of Peru, it is known from the cave Baño Nuevo-1 at 750 masl 80 km, northeast of Coyhaique, Chile [73].

This genus is a small-sized sloth and exhibits a peculiar mosaic of cranial and postcranial characters, some of which are considered convergent with other sloths. These features include five upper and four lower quadrangular teeth, characteristic of Megatheriidae). While originally placed in the Megalonychidae, this genus is distinguished from all other members of the family as the first upper and lower tooth are not modified into a caniniform and are separated from the molariform teeth by a diastema. Currently, it can only be considered a megatheroid sloth. The post-cranial skeleton indicates potential climbing capabilities. An arboreal mode of life of *D. nordenskioldi* was postulated based on the considerable mobility of the elbow, hip, and ankle joints, a posteriorly convex ulna with an olecranon shorter than in fossorial taxa, a radial notch that is oriented more anteriorly than in other fossil sloths and forms an obtuse angle with the coronoid process thus increasing the range of pronation–supination, a proximodistally compressed scaphoid, and a wide range of digital flexion [74]. The femur is not known so there is no estimate of body mass.

Analysis of pollen and spores from dung, possibly from a large sloth from Trigo Jirka, which also had remains of *Diabolotherium*, documented over 20 species of plants. Most of the pollen is of *Alnus* sp., the Andean alder (Betulaceae). *Alnus acuminata* is considered the only *Alnus* species in the Andes [75]. Its distribution ranges from Southern Mexico (~17° N latitude) to Northwest Argentina (~28° S latitude) and grows in the mountainous areas of Central America and the Andes. In Ecuador and Peru, *A. acuminata* occurs from 1500 m to ~3400 m elevation and is found in montane cloud forests.

Fossil pollen records from the high Andes of Ecuador and Northern Peru show that *Alnus* increased in abundance following the warming associated with the onset of the Holocene. *Alnus* pollen abundance peaked between 9000 and 5000 years BP. However, between 6500 and 4500 years BP, the abundance of *Alnus* pollen declined markedly both within and above its modern distribution range [76]. Spores of club moss (*Lycopodium*), ferns, and fungi were also recovered from the dung. Indications of the local flora during the Pleistocene are not as well preserved in the other two caves. In Jatun Uchco analysis of pollen and spores indicates cacti were the most common plant of the region at the time of deposition and Amaryllidaceae and Thymelaceae were also present. No pollen was found in sediment samples from Cueva Roselló. However, spores of the club moss *Lycopodium* were common [72]. The radiocarbon date of the *Diabolotherium* from Trigo Jirka was 29,140 ± 260 BP.

The presence of Andean alder, which reaches 20–25 m in height and is associated with montane cloud forest may have provided the habitat for which *Diabolotherium* was adapted given its proposed arboreal habits and that most records are from high elevations except for one.

### 4.2.8. *Ahytherium aureum*—Brazilian Intertropical Region (Figure 3C)

The type locality and only known site for this taxon is the cave system, Poço Azul, in Chapada Diamantina National Park, Bahia Brazil. The relative muzzle width index and the occlusal surface area *Ahytherium aureum* indicate was adapted to a mixed-feeder diet, varying in the consumption of C3 and C4 plants [23]. Based on femur length the body mass is estimated at 391 kg.

### 4.2.9. *Australonyx aquae*—Brazilian Intertropical Region (Figure 3D)

This megalonychid is currently only known from the cave system, Poço Azul, in Chapada, Diamantina National Park, Bahia Brazil. The relative muzzle width index and the occlusal surface area of *Australonyx aquae* indicate it was adapted to a mixed-feeder diet, varying in the consumption of C3 and C4 plants [23]. The femur is not known for this taxon so there is no estimate of body mass, but the skull is about half the size of *Ahytherium* so it is a small taxon.

### 4.3. Caribbean Megalonychids

Late Pleistocene records of sloths are known from Cuba, Hispaniola, and Puerto Rico [77,78]. All Caribbean sloths are members of the Megalonychidae and are represented by four genera and multiple species. The most recent review of the taxonomic diversity is White and MacPhee [78], but subsequently to that review, additional species have been proposed [79,80]. While the body mass of the Caribbean sloths is significantly smaller than their mainland counterparts, many of the species are significantly larger than those of the next largest mammalian group, rodents, so the sloths constitute the "mammalian megafauna" on these islands. The extinction of the Caribbean sloths occurred much later, ca. 4000 to 5000 years BP than those on the continent [5]. Given their geographic isolation, evolutionary history in isolation, and different ecological requirements, which may have been distinct on different islands, as well as the lack of mammalian predators, the causes of their subsequent extinction are most likely quite different from that of their relatives on the mainland.

The largest genus is *Megalocnus* with two species, *M. rodens* (Figure 4A) in Cuba and what was originally described as *M. zile* from Haiti. *Megalocnus zile* is now considered a junior synonym of *Parocnus serus* [80]. The Cuban record of the genus is extensive, and it has been recovered from multiple caves and from spring deposits, in contrast to *M. zile* which is only known from the holotype. Based on femur length the body mass of *M. rodens* is estimated at 146 kg.

Closely related to *Megalocnus* is *Parocnus* with three species, *P. serus* (Figure 4B) from Haiti, *P. browni* in Cuba, and *P. dominicanus* from the Dominican Republic [78,80]. The size varies among the three species and body mass estimates based on femur length are 79, 42, and 32 kg, respectively.

Originally described based on specimens from Puerto Rico *Acratocnus* includes four species, *A. odontrigonus* (Puerto Rico) (Figure 4C), *A. ye* (Figure 4D), *A. simorhynchus* (Hispaniola), and *A. antillensis* (Cuba), *A. odontrigonus* and *A. ye* have similar estimates of body mass of 15 kg based on femur length while *A. antillensis* is smaller with an estimated 10 kg.

Currently, *Neocnus* includes five species, *N. major* and *N. gliriformis* on Cuba and *N. comes* (Figure 4E), *N. Dousman*, and *N. toupiti* on Hispaniola. The estimated body mass of *N. comes* is 8.36 kg. Based on an analysis of the feeding mechanics of *Neocnus* it is interpreted as a selective feeder and had a folivorous diet [81].

*Paulocnus* is known only from the island of Curaçao in the Netherland Antilles off the coast of Venezuela. As the geologic age of the material is not known and may not be part of the late Pleistocene fauna, it is not considered further.

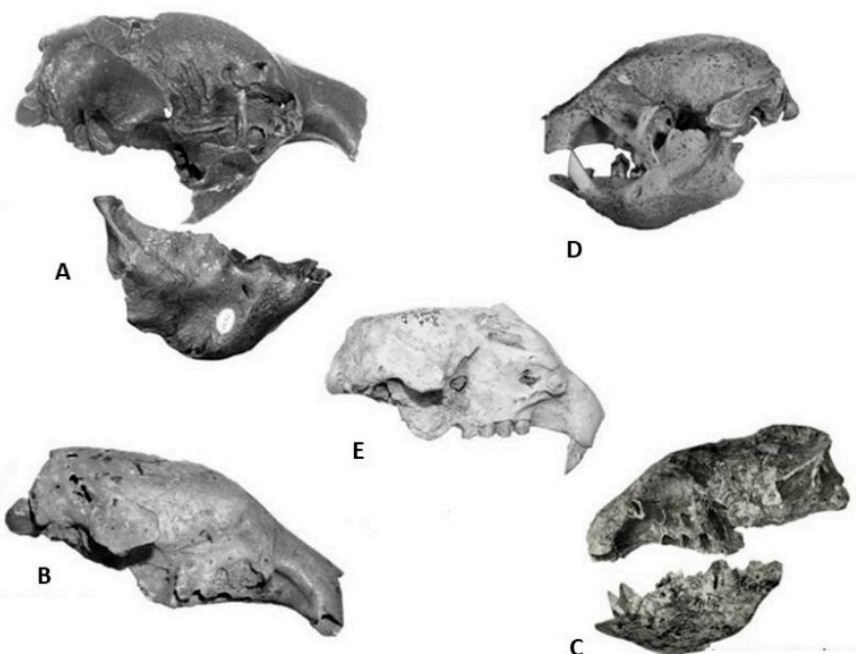

**Figure 4.** Family Megalonychidae. Caribbean taxa. *Megalocnus rodens* (**A**), *Parocnus serus* (**B**), *Acratocnus odontrigonus* (**C**), *Acratocnus ye* (**D**), *Neocnus comes* (**E**).

Most studies of the Caribbean sloths have focused on their taxonomy and relationships with other members of the family [78]. White [82] proposed that *Neocnus* and *Acratocnus*, the two smaller genera of Caribbean sloths, may have been arboreal or at least semiarboreal in the locomotion-based morphological indices of their limb bones. Coprolites considered to be from sloths have been reported from three localities in Cuba. Coprolites are abundant at Solapa del Megalocnus (Cueva del Megalocnus) and occur throughout the deposit. The coprolites are attributed to either *Megalocnus rodens*, *Parocnus browni,* or *Miocnus antillensis* (*Miocnus* is now placed in *Acratocnus*). A coprolite from Baños de Ciego Montero identified as sloth contained pollen grains from the Maranthaceae (arrowroot), Convolvulaceae (bindweeds or morning glories), and Dioscoreaceae (yams) [83,84]. An illustrated rounded end of a cylindrical coprolite from Quemado de Güines attributed to a sloth that had pollen of the legume *Canavalia* [84].

### 4.4. Family Mylodontidae
#### 4.4.1. *Mylodon darwinii*—Pampas and Patagonia (Figure 5A)

In the narrow muzzled *Mylodon darwinii,* the premaxillae are completely fused to the maxillae and have a strong anterior ascending process that fused with a descending process of the nasals, forming an unusual complete arch [85]. In narrow-muzzle forms, such as *Mylodon darwinii* the muzzle is part of a robust structure that, combined with the fused symphysis of the mandible, provides a functional adaptation to resist considerable stress, suggesting that the processing of favored plant materials required significant effort. In *M. darwinii* at least, the degree of fusion of the premaxilla and maxilla, the stout nasal arch, and the loss of the first upper tooth suggest the presence of a horny structure on the premaxilla, analogous to the premaxillary pads in bovids, that would aid in clipping or tearing off the food. Narrow-muzzled sloths such as *Mylodon* are interpreted as mixed or selective feeders, with a prehensile lip that was used to select specific plants or plant parts [86].

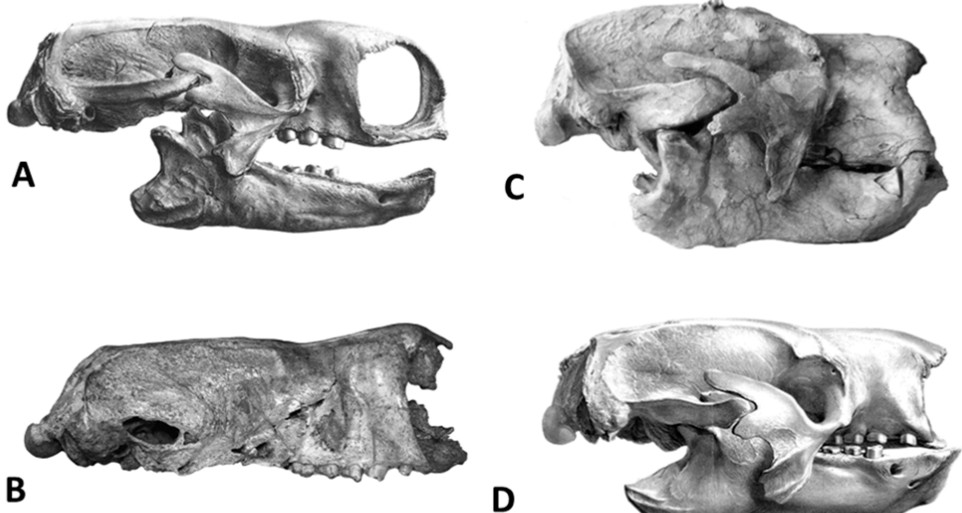

**Figure 5.** Family Mylodontidae (Mylodontinae). *Mylodon darwinii* (**A**), *Glossotherium tropicorum* (**B**), *Lestodon armatus* (**C**), *Paramylodon harlani* (**D**).

The body mass estimate is estimated as 1986 kg [17]. Brambilla and Haro [87] examined postcranial elements of *Mylodon* and proposed that *Mylodon listai* from Patagonia was significantly different from *Mylodon* from the Pampean region which has been referred to as *Mylodon darwinii*. They interpreted these anatomical features as indicating *M. listai* had less bowed legs than *M. darwinii* and was an adaptation to a less fossorial lifestyle in their habitats in Southern Patagonia and not indicative of differences in body mass in the southern specimens, as *M. listai* reached similar or even larger sizes than *M. darwinii*.

The distribution of *Mylodon* extends from Mojotorillo, Tomina Province, Bolivia (19.6° S) to Fells Cave, Cerro Sota, Pali Aike, Chile (52° S). Its distribution is primarily on the eastern side of the Andes with most localities in Argentina [88] with two in Bolivia, two in Paraguay, one in Uruguay, and three in Chile, except for two records in Chile from Los Vilos, on the coast of Choapa Province and Baño Nuevo, XI Region, Chile at an elevation of 750 m [89]. Although this distribution may encompass either one or two species, until further study can establish specific identifications at each locality, at this time all late Pleistocene records of the genus are attributed to a single species, *Mylodon darwinii*.

Based on plant fragments in the dung of *Mylodon*, its diet included grasses and sedges along with other plant species associated with modern cool, wet sedge grasslands [90]. Fossil pollen assemblages of *M. darwinii* were also recovered from coprolites from the Cueva del Milodon [91,92]. The feces dated between 13,470 ± 180 and 10,575 ± 400 rcy BP. The pollen assemblages were dominated by Poaceae (grass steppe) and *Empetrum* (tundra or dwarf scrub heath) and fed on a diet that included Cyperaceae, Poaceae, and species including *Marsippospermum grandiflorum*, *Plagiobotrys albiflorus*, and *Oreobolus obtusangulus*. These species nowadays occur in the open, cool, wet sedge grasslands of Western Patagonia. *Empetrum rubrum* occupies a wide range of open habitats, from dry steppe to Magellanic moors. The presence of substantial amounts of *Fragaria* and *Azorella* pollen in the feces may indicate that *Mylodon* was also able to select and consume specific plants and suggests *Mylodon* could also be regarded as a selective feeder [93]. Analysis of pollen preserved in the dung of *Mylodon* from Cueva del Milodon confirms earlier studies that *Mylodon* was a grazer. However, based on isotopic analyses of nitrogen of specific amino acids, it has been suggested that Darwin's ground sloth *Mylodon darwinii* may have also been an opportunistic omnivore [94]. *Mylodon darwini* has been traditionally linked to open areas and temperate to cold semiarid climates [95,96]. *Mylodon* appears to have had great ecological tolerance and was capable of inhabiting climates ranging from cold and arid to warm and humid, and even montane environments. Radiocarbon-dated dung has produced dates around 13,140 ± 55 BP (15,927–15,522 calendric years), which fits with other radiocarbon dates showing the early

late Glacial presence of *M. darwinii* in the vicinity of Última Esperanza at the southern tip of South America. Contemporaneous oxygen isotope data from the Antarctic Dome indicates that *Mylodon* lived this far south during a warming phase of the late Glacial, ca. 1000 years before the start of the Antarctic Cold Reversal [93].

4.4.2. *Glossotherium robustum*—Pampas and Patagonia

The taxonomic history of *Glossotherium* is quite convoluted especially that of *G. robustum* and the genus has served as a catch-all dumping ground for many different mylodontid taxa such as *Mylodon robustus*, *Mylodon gracilis*, *Pseudolestodon lettsomi*, *Pseudolestodon myloides*, *Pseudolestodon morenoii*, and *Glossotherium wegneri* [97] with continuing disagreement on the genus of this last species as discussed elsewhere. Mones [98] listed 17 species that had been included in *Glossotherium*. This taxonomic history has impacted our understanding of the distribution and paleoecology of this species. The track generated by Gallo et al. [11] for *Glossotherium robustum* extends across most of South America north of the Pampas. Consequently, records historically attributed to this species, but now recognized as distinct species such as *G. tropicorum* and *G. wegneri* (or *Oreomylodon*) in Ecuador and *G. phoenesis* in Brazil were included in the analysis and thus have clouded our understanding of the distribution of this taxon. Although their results may not be applicable to the species, they do provide some insight into the distribution of the genus. While the taxonomy of sloths from the late Pleistocene is relatively consistent to genus, the number of valid species is more nebulous and indicates the need for detailed systematic studies at the species level to provide a firmer foundation for understanding their ecology, distribution, and extinction.

Bargo [99] concluded that *Glossotherium robustum* was a bulk feeder and the wide muzzle would have allowed the animal to procure substantial amounts of food, and its muzzle morphology was the best-adapted of all ground sloths to a grazing niche [86]. Morphologic and biomechanical analyses of the late Pleistocene mylodontids indicate that hypsodonty was unlikely to be due solely to feeding behavior, such as grazing [100]. Some mylodontids including *Glossotherium robustum*, had a forelimb morphology indicating they were capable diggers that likely dug for food. They would have potentially ingested abrasive soil particles which also may have played a role in shaping their dental characteristics. Among the morphological features present in *Glossotherium* associated with digging, is the large and elongated olecranon process of the ulna, higher strength indicators of the humerus, and the large, flat, wide, and straight ungual phalanges [101]. Related to digging adaptations, the presence of claw marks on the walls and roof of sedimentary structures near Mar del Plata are attributed to palaeoburrows built by fossil animals including sloths such as *Glossotherium* [102]. Postulated uses of these burrows include escaping predation or avoiding alternatively excessively cold or warm climatic conditions and conserving energy and water. During the early Pleistocene (Ensenadan) the climate of northern Argentina was warmer than present [103] but by the last part of the Pleistocene (Lujanian) it had shifted to a colder, dry steppe [104]. As in the living xenarthrans the extinct sloths are considered to have had low body temperatures, low basal rates of metabolism, and high thermal conductance [105]. In this later environment, the mylodontids may have needed a warmer and humid place to breed, use as a den site for young or even to survive, during the colder season. The estimated body mass is between 1200 kg and 1700 kg [101].

The results of stable isotope analysis for *Glossotherium robustum* from the late Pleistocene La Postrera Formation in partido de Lobería, Buenos Aires Province, Argentina averaged at –20.5‰ for the $\delta^{13}$C and +10.2‰ for the $\delta^{15}$N. The results of $\delta^{13}$C for *G. robustum* indicate a preference for C3 vegetation in open environments and it was a bulk feeder. This data complements its inferred trophic habits based on morphological and biomechanical studies [106].

The results of the $\delta^{15}$N for *G. robustum* could reflect either its non-ruminant herbivorous physiology of sloths or the colder and drier climate inferred for the Pampean region in the late Pleistocene, than at present. The age of the *Glossotherium* is ca 12,000 years BP placing it around the Younger Dryas (circa 12,900 to 11,700 years BP) when the region was colder and drier than

the present [104,107–109]. The cold climatic phase is considered to have terminated about 8500 years BP, with a shift in the climate to humid subtropical conditions [107]. Coinciding with this shift in climate was a transition from the psammophytic scrubby steppe to a humid phase [32], with swamps in the northeast of the Pampas (Cerro La China, Empalme Querandíes, and La Horqueta II) beginning to form around 11,000 years BP. (104,109). Tonni et al. [109] proposed the humid conditions may have corresponded to local conditions in this area, whereas in the rest of the Pampas, there was deposition of aeolian sediments under arid conditions as is recorded in the La Postrera Formation.

### 4.4.3. *Glossotherium tropicorum*—West Coast South America (Figure 5B)

*G. tropicorum*, (Figure 5D) was originally described from the late Pleistocene tar seeps of La Carolina, Santa Elena Peninsula, Ecuador [110]. The species is also known from the late Pleistocene tar seep localities of Corralito (Ecuador) and Talara (Peru). *Glossotherium tropicorum* is similar in size to *Glossotherium robustum* and *Oreomylodon wegneri*. It is one of the two mylodontine sloths present in the northwestern part of South America in the late Pleistocene, the other is *O. wegneri*. While both taxa are present in the general region, their individual distributions did not overlap, and the two species have not been recovered together in a fauna. The distribution of *G. tropicorum* is restricted to the lowland coastal areas, with a range of elevation of 0–600 m while *O. wegneri* is found inland and in the highland areas of the Andes at elevations of 2450–3100 m [110,111]. No stable isotope analysis has been conducted for either species so there are no specifics of the ecology of each species. Based on the differences in their cranial morphology and non-overlapping distributions it is highly probable that they also had distinctive and different ecologies. While both were most likely to have been grazers, or mixed feeders, the plants consumed by each species would have certainly been different given the difference in elevation of their respective ranges.

### 4.4.4. *Glossotherium phoenesis*—Brazilian Intertropical Region

This species of *Glossotherium* is known from the type locality, Toca dos Ossos, and Ouro Branco, in Bahia, Brazil, and in the state of Minas Gerais so is found only in the BIR. [112].

The relative muzzle width index and the occlusal surface area of *Glossotherium phoenesis*, adapted to a mixed-feeder diet, vary in the consumption of C3 and C4 plants [23]. The estimated body mass is around 1041 kg based on the average femur length of three individuals.

### 4.4.5. *Lestodon armatus*—Pampas and Patagonia (Figure 5C)

*Lestodon* is the third largest of the extinct ground sloths after *Eremotherium* and *Megatherium* with an estimated body mass of 3397 kg. The highest number of localities is in Buenos Aires Province and Northeastern Argentina into Uruguay with a few records in Paraguay and some scattered outliers in Brazil. The localities range in elevation from 0–900 m [113,114]. The biogeographic track for *Lestodon* [11] includes records in Northwestern Brazil which suggest its presence in more forested tropical environments compared to its habitat in northern Argentina. A reevaluation of the specimens assigned to *Lestodon* from this region is warranted.

The average stable isotope values for *Lestodon armatus* from Arroyo del Vizcaíno, Sauce, Department of Canelones, Uruguay was –18.8‰ for the $\delta^{13}$C and +9.5‰ for the $\delta^{15}$N. The results of $\delta^{13}$C for *L. armatus* indicate a preference for C3 vegetation in an open environment. This is congruent with the trophic habits inferred for these ground sloths based on morphological and biomechanical studies that suggest they were bulk feeders [106].

### 4.4.6. *Paramylodon harlani*—Temperate North America (Figure 5D)

While originally interpreted as a grazer, based on its skull morphology and dentition Naples [115] proposed that *Paramylodon harlani* (then referred to *Glossotherium*) may have been more of a mixed feeder. The body form and limb structure are comparable to *Glossotherium* and *Mylodon*. The estimated body mass is 1392 kg. The range of results for stable

isotope values obtained from different localities suggests dietary flexibility in *Paramylodon* and it utilized a variety of different open environments, from grasslands to more wooded areas but did not utilize closed canopy environments. This is also seen in other localities as well based on the reconstruction of the dietary habits of other herbivorous mammals from El Cedral, La Cinta-Portalitos, and Villaflores in Mexico, which indicate *Paramylodon* lived in areas with heterogenous vegetation dominated by grasslands [116,117]. There is a close association of *Paramylodon* with other grazing herbivores [118].

Stable isotope analysis of *Paramylodon* has provided an independent means to assess its diet. *Paramylodon harlani*, from the Gulf Coast of Texas (Ingleside fauna), showed $\delta^{13}$C values of near –4‰, in the range expected of mixed feeders, but close to the carbon isotopic composition of modern and fossil grazers [119]. Specimens of *Paramylodon* from Rancho La Brea sampled for stable isotopes had values with a $\delta^{13}$C range between –19.99 and –21.49 (average –20.99). The $\delta^{15}$N values ranged from +6.44 to 10.12 (average 7.93) [120]. Two individuals from the Willamette Valley, Oregon had $\delta^{13}$C values of –20.8 and –21 and $\delta^{15}$N values of +7.4 and 6.6 indicating they were feeding primarily on C3 vegetation in a paleoenvironment of open grassland and sparse canopy [121]. The inferred diet of *P. harlani* from Valsequillo, Mexico based on $\delta^{13}$C indicated this individual was primarily a grazer [116]. While two samples from Térapa, Sonora produced quite different $\delta^{13}$C values of $-6.3$ and $-0.5$ the values are comparable to those of other herbivore taxa from the site and support the interpretation of a mosaic of habitats that contained a variety of herbivores with different dietary preferences. Based on the sloth and other herbivores there is no isotopic evidence to suggest the presence of closed canopy forests or the strict consumption of C3 plants by any of the mammalian herbivores found at Térapa. Most of the herbivores were exploiting a wide range of vegetation within a tropical dry forest and grassland. The sloth had $\delta^{18}$O values of $-6.3$ to $-4.9$ and based on the oxygen isotope values of the other herbivores it does not appear there was any seasonal variation in the $\delta^{18}$O values (~4‰) compared to modern desert environments of the area which display significant annual variations in $\delta^{18}$O.

### 4.4.7. *Oreomylodon wegneri*—Andes and Altiplano

*Oreomylodon wegneri* is frequently found in the north and central parts of the inter-Andean valleys of Ecuador, between 2450 and 3100 m above sea level in sediments that have been assigned to the late Pleistocene. It is known primarily from three localities, Quebrada Pistud, Carchi Province; Quito's Valley, Pichincha Province; Quebrada Chalan, Chimborazo Province.

The high-elevation habitat of *O. wegneri* exceeds that of all the other mylodontines from the late Pleistocene. The distinctive enlargement of the nasal region and the nasal atrium could be an adaptive feature to aid in regulating body temperature and for the filtration and humidification of cold and dry air in the Ecuadorian Andes during the late Pleistocene [111]. During the late Pleistocene, the snow line in the northern and central Andes would have been 1000 m lower than the current limit [122,123] with terminal moraines at 3500 m [124]. The Cangagua Formation, which occurs between 1900 and 3000 m contains the majority of *O. wegneri* remains and would have been deposited under these conditions. The estimated body mass is 751 kg based on the average of the smallest and largest femora reported [110].

### 4.4.8. *Ocnotherium giganteum*—Brazilian Intertropical Region

Previously included as a subgenus of *Glossotherium*, *Ocnotherium* is now considered a distinct genus. Records of *Ocnotherium* are limited and no complete skeletons are known. An associated radius and complete left manus from Gruta dos Brejões referred to *Ocnotherium* are among the better-preserved specimens of the genus [125]. Subsequent studies considered *Ocnotherium* to be morphologically close to *Lestodon* [9]. The estimated body mass is 842 kg [23].

The distribution of *Ocnotherium* is limited to the BIR from localities in the states of Bahia, Rio Grande do Norte, and Minas Gerais. Based on the latitudinal locations of the

few sites known for this sloth, its diet consisted primarily of C4 gramineae species, but there is no stable isotope data to independently confirm its diet. Cranial material does permit the calculation of the relative muzzle width index and the occlusal surface area of *Ocnotherium* which indicates it was a mixed feeder with a varying consumption of C3 and C4 plants [23].

### 4.4.9. *Mylodonopsis ibseni*—Brazilian Intertropical Region

The initial description of *Mylodonopsis ibseni* [126] considered it to be morphologically close to *Mylodon darwini*. Bargo et al. [26,99] assigned a generalist diet to *M. darwini* and tentatively attributed a similar diet to *M. ibseni*. The calculated relative muzzle width index of 0.53 placed *Mylodonopsis* in the mixed feeder category but the occlusal surface area could not be calculated [23].

As with *Ocnotherium*, little is known about *Mylodonopsis*. There is no stable isotope data and no estimates of body mass, although it is similar in size to *Mylodon darwinii*. It is currently only known from two sites in the BIR, Gruta dos Brejões and Toca dos Ossos, Bahia state, Brazil.

### 4.4.10. *Scelidotherium leptocephalum*—Pampas and Patagonia (Figure 6A)

In the narrow muzzled *Scelidotherium leptocephalum* the premaxillae are more elongated and completely fused to the maxillae, unlike *Catonyx* where they are shorter and unfused. The estimated body mass is 1119 kg [17].

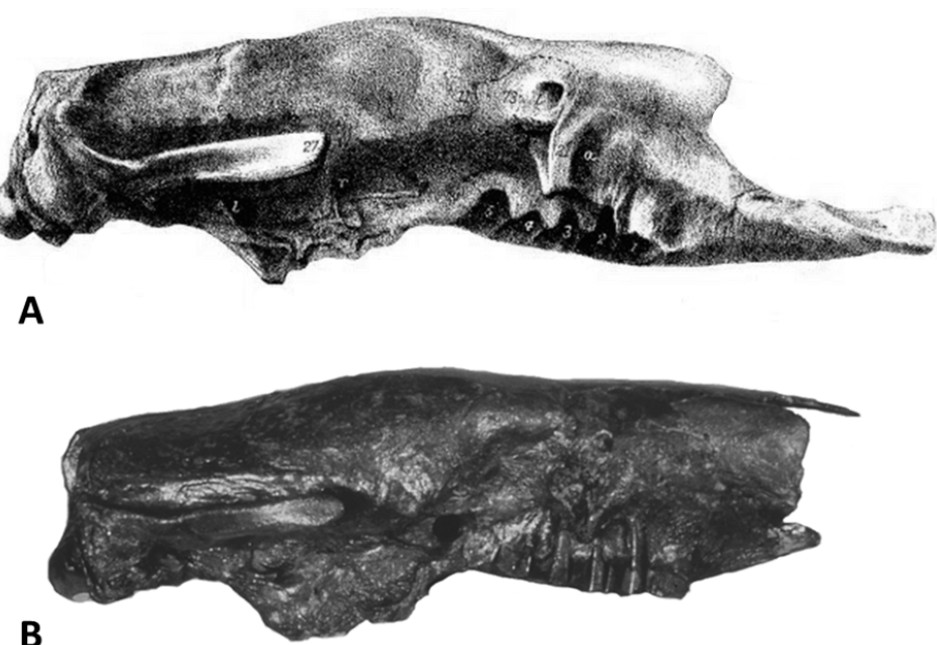

**Figure 6.** Family Mylodontidae (Scelidotheriinae). *Scelidotherium leptocephalum* (**A**), *Catonyx chiliense* (**B**).

The $\delta^{13}$C values of *Scelidotherium leptocephalum* from Santa Rosa, La Pampa Province and Playa del Barco, Buenos Aires Provinces in Argentina were −3.9 and −3.2‰ and the $\delta^{18}$O$_{CO3}$ were 26.8 and 29.7 [28]. The $\delta^{13}$C values from Santa Rosa indicate an intermediate C3–C4 diet while the $\delta^{13}$C values at Playa del Barco indicate a general preference of intermediate $C_3$–$C_4$ open vegetation. Based on $\delta^{18}$O values of these two localities the sloth tolerated the environmental changes related to the transition from glacial to interglacial conditions phases with the reduction of subtropical and tropical biomes and the expansion of open biomes, related to glacial periods.

The distribution of *Scelidotherium leptocephalum* was a well-defined region encompassing Southern Uruguay, Northern Argentina, and Southern Paraguay with its eastern border on the coast, and western border on the flanks of the Andes [11].

### 4.4.11. *Catonyx cuvieri*—Brazilian Intertropical Region

The type specimen of *Catonyx cuvieri* comes from Lapa Grande de Genette, Minas Gerais State, Brazil. Lund described multiple other genera and species from other caves in the vicinity of Lagoa Santa, but many of these are now considered junior synonyms of *C. cuvieri*. The estimated body mass is 541 kg based on an average of four femora.

The primary range of *C. cuvieri* is the BIR [127] and it has since been reported from other sites, gruta da Marota, Bahia [128], Fazenda Charco, Sergipe [129]; João Cativo, Itapipoca, Ceará [130]; São Raimundo Nonato, Piauí [131]; Igaci, Algoas [132]; Poço Azul, Nova Redenção, Bahia [133]; south to Abismo Iguatemi Cave, Iporanga and the upper Ribeira Valley, São Paulo State [134] and Uruguay [16].

Marota Cave (Andaraí municipality, Bahia, Brazil) is in the karst system of the Una-Utinga basin, Chapada Diamantina in the BIR. *Catonyx cuvieri* is present in the fauna. The $^{14}$C AMS dates are around 11,000 years BP for all the studied taxa, placing the records in the late Pleistocene. The carbon isotopic value for *C. cuvieri* is −11.19‰, indicating it was a dietary specialist, feeding primarily on C3 plants (piC3 > 80%). The information obtained from ($\delta^{18}$O) varying from 4.06‰ to −0.08‰ ($\mu\delta^{18}$O = −1.94‰), associated with $\delta^{13}$C data, suggests a forested environment in the region around 11,000 years ago [128].

### 4.4.12. *Catonyx cuvieri*—Pampas, and Patagonia

The only records of *Catonyx cuvieri* in the Pampas are from the late Pleistocene Dolores Formation in the Departments of Soriano, Colonia, and Canelones, Southern Uruguay [16]. The Dolores Formation is interpreted as having been deposited during the LGM under more arid climatic conditions than at present [135,136]. Fragments of wood of *Prosopis* sp.(algarrobo) and *Salix humboldtiana* (sauce criollo) have been found in the formation. This is the southernmost record of these species at ca. 34.5° S as the next closest record is to the north in the state of Rio Grande do Sul, Brazil ca. 30.5° S. The vertebrate fauna of the Dolores Formation includes taxa interpreted as indicative of a dry and cold climate [137]. This suggests *Catonyx cuvieri* was not as narrow in its ecological requirements as previously thought and could adapt to colder and drier climatic conditions.

### 4.4.13. *Catonyx chiliense*—West Coast South America (Figure 6B)

The type locality for *Catonyx chiliense* is Tamarugal, Tarapacá Province, Chile. It is also known from Santiago, Chile, the tar deposits on the Santa Elena Peninsula Ecuador, and Talara and Pampa de los Fósiles, Cupisnique, Peru, and Ulloma, Bolivia. Recently a scelidothere identified as *Scelidodon chiliense* was reported from the east side of the Andes from the late Pleistocene—early Holocene Uspara Formation of San Luis Province, Argentina [138] which extends the range of the species.

Based on the distribution of this species its preferred habitat is drier or more arid environments, so was quite different from that of *C. cuvieri* from the BIR. The skull morphology and dentition of the two species are also generally similar so the interpretation of their feeding habits is as broadly similar as generalist feeders, although the vegetation of the two different regions would have been very different. There are no studies of the stable isotopes for the diet of this species. The body mass estimate is 697 kg based on an average of six femora.

### 4.4.14. *Valgipes bucklandi*—Brazilian Intertropical Region

The relative muzzle width index and the occlusal surface area of *Valgipes bucklandi*, indicates it was adapted to a mixed-feeder diet, varying in the consumption of C3 and C4 plants [23]. The estimated body mass is 724 kg based on femur length.

This Pleistocene ground sloth is primarily found within the BIR in the states of Bahia, Minas Gerais, Piauí, and Rio Grande do Norte. Outside the BIR, it has been reported from Mato Grosso do Sul and at Arroyo del Vizcaíno, Canelones, Uruguay [139].

Analyses of stable carbon isotopes confirm a browsing diet for *V. bucklandi* based on $\delta^{13}$C (−10.17‰,) and are tentatively attributed to the same diet and habits as for the *C. cuvieri*. [140].

*Valigipes* from the BIR were considered to have a narrow niche overlap with other herbivores in the region that fed principally on C4 plants (>70%; O = 0.24–0.43) [140]. The only specimen of *V. bucklandi* from Felipe Guerra, Rio Grande do Norte had a diet primarily composed of C3 plants from open landscapes (94%; $\delta^{13}C = -10.2‰$; $\delta^{18}O = -1.7‰$), which reflects a narrow niche breadth. Stable isotope data for the individual from Uruguay for $\delta^{15}N$ was 12.66‰. The percentage of carbon present in the sample was 2.25% and the $\delta^{13}C$ value was −21.13‰. The carbonate sample showed a $\delta^{13}C$ value of −11.76‰ and a $\delta^{18}O$ value of −8.09‰. The $\delta^{13}C$ carb-coll was 9.37 [140]. In contrast, the results of the isotopic analysis of the individual from Arroyo de Vizcaíno in Uruguay show differences in the obtained $\delta^{18}O$ value (−8.9‰) when compared with the value previously obtained from a specimen from the BIR (−1.7‰), indicating a colder and more humid environment [140]. Stable isotope analyses of *V. bucklandi* from Uruguay ~2000 km south of its distribution in Brazil produced $\delta^{13}C$ values ($\mu\delta^{13}C$ of −10.2) consistent with those of the BIR, indicating a similar preference for C3 plants. In contrast, the results of $\delta^{18}O$ values indicated colder and more humid environments, clearly different from those obtained from BIR specimens.

*V. bucklandi* could have preferred more tropical closed environments. In fact, an open mosaic habitat with patches of mixed vegetation typical of more closed environments was proposed for the Arroyo de Vizcaíno site based on the presence of other browser taxa in addition to some of the sloths. These results support the existence of mixed faunas composed of southern and northern taxa coexisting in Southern Uruguay at the onset of the LGM and provide evidence of the existence of an open mosaic habitat with patches of mixed vegetation in the region.

### 4.5. Mylodontid Indet

Mylodontid material has been reported from Southern Mexico–Central America, but usually as casual mentions with no details such as the records from Guatemala [141] or based on limited material such as the mylodontid ectotympanic from Actun Spukil on the Yucatan Peninsula [142]. A mylodontid mandible from El Hatillo, Panama was initially identified as cf. *Glossotherium tropicum* [143] but has since been referred to *Paramylodon harlani* [144].

An indeterminate mylodontid is also known from a tar deposit, El Breal de Orocual, Venezuela [43].

### 4.6. Family Nothrotheriidae

4.6.1. *Nothrotherium maquinense*—Brazilian Intertropical Region, Pampus and Patagonia (Figure 7A)

*Nothrotherium maquinense* is the smallest of the late Pleistocene ground sloths with an estimated body mass of 210 kg [51]. It is about two-thirds the size of its North American counterpart, *Nothrotheriops*. Overall, the skeleton of *Nothrotherium* is more gracile than that of *Nothrotheriops*. A detailed overview of differences in the major bones of the skeleton of the two genera is provided by de Paula Couto [145]. A complete skeleton was recovered from Gruta dos Brejões, State of Bahia, Brazil [146].

The type of *Nothrotherium maquinense* comes from Lapa Nova de Maquin6, state of Minas Gerais, Brazil, and a second species recovered from another cave in the area, Lapa de Escrivania n° 5 was described as *N. escrivanense*. Most workers consider there to only be a single species, but Pujos [147] considers both species valid. *Nothrotherium* has been recovered from multiple caves in the BIR in the states of Bahia, Rio Paraíba, and Ceará south to the state of São Paulo [11,127]. Outside of Brazil *Nothrotherium* cf. *N. maquinense* was reported from the late Pleistocene Sopas Formation of Uruguay [148]. The Sopas Formation was deposited during Marine Isotopic Stage 3. Plant taxa in the formation indicate open habitats, savannahs, and woodlands including gallery forests along perennial river systems with mostly tropical to temperate climates although some taxa suggest arid to semiarid environments [149].

The vertebrate fauna from Marota Cave (Andaraí municipality, Bahia, Brazil includes two sloths, *Catonyx cuvieri* and *Nothrotherium maquinense* [128]. The carbon isotopic value for *N. maquinense* was −12.12‰. These data suggest *N. maquinense* was a dietary specialist

feeding more on C3 plants (piC3 > 80%). The information obtained from $\delta^{18}O$ varied from $-4.06‰$ to $-0.08‰$ ($\mu\delta^{18}O = -1.94‰$), which combined with $\delta^{13}C$ data, suggests a forested environment in the region around 11,000 years ago.

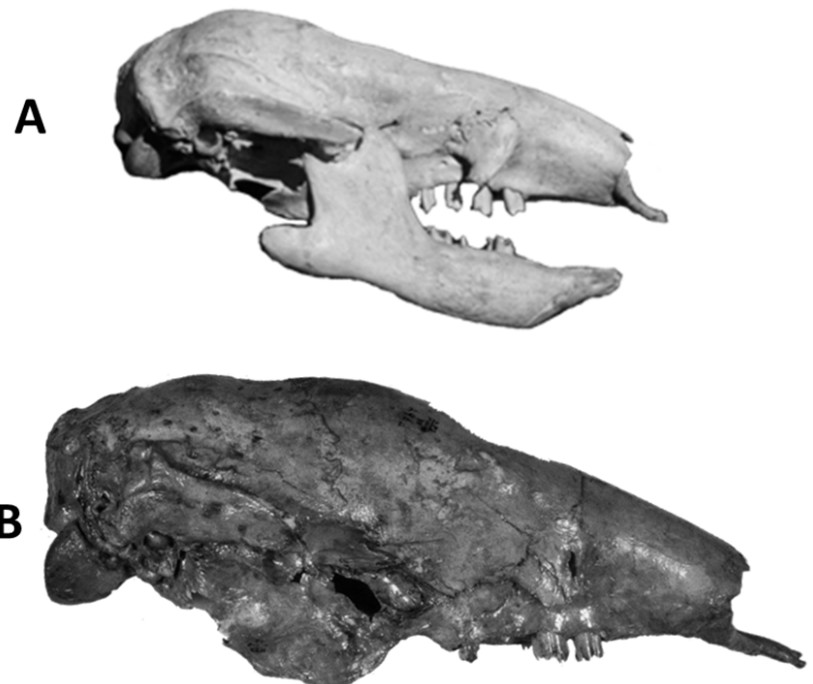

**Figure 7.** Family Nothrotheriidae. *Nothrotherium maquinense* (**A**), *Nothrotheriops shastensis* (**B**).

4.6.2. *Nothrotherium escrivanense*—Brazilian Intertropical Region

*Nothrotherium escrivanense* was reported from Argentina [150].

Currently, there is no agreement regarding the validity of *Nothrotherium escrivanense* [145,146] and others consider *Nothrotherium escrivanense* a junior synonym of *Nothrotherium maquinense*. However, Pujos [147] considers *Nothrotherium escrivanense* a valid species. A systematic review of the Pleistocene Nothrotheriinae would shed light on the diversity of this group in the late Pleistocene of Argentina.

4.6.3. *Nothrotheriops shastensis*—Temperate North America, Southern Mexico, and Central America (Figure 7B)

*Nothrotheriops shastensis* is the smallest of the late Pleistocene ground sloths in temperate North America with an estimated body mass of 463 kg.

*Nothrotheriops* is the most xeric adapted of all the sloths in North America based on multiple localities in the Southwestern United States, many of which are dry caves that have preserved soft tissues as well as the animal's dung. This is the only sloth species in North America with preserved dung as confirmed by DNA fingerprinting [151]. Analysis of macrobotanical remains in the dung has permitted a detailed determination of the diversity of plants in its diet [152]. Many of the taxa are restricted to desert environments as indicated by plant taxa including desert globemallow (*Sphaeralcea ambigua*), Nevada Mormon tea (*Ephedra nevadensis*), saltbushes (*Atriplex* spp.), catclaw acacia (*Acacia greggii*), creosote (*Larrea* sp.) and yucca (*Yucca* spp.). The dung also included many other plants with wider distributions that are found in other habitats as well. Examples include common reed (*Phragmites communis*), Utah juniper (*Juniperus utahensis*), cattail (*Typha* sp.), three-leaf sumac (*Rhus trilobata*), and gray rabbitbrush (*Ericameria nauseosa = Chrysothamnus speciosus*). Similar suites of plants have been identified in the dung of *Nothrotheriops* from Gypsum Cave, Nevada [153] and Shelter Cave, New Mexico [154]. Gas chromatography and mass spectrometry of coprolites of *Nothrotheriops* indicated a dominance of epismilagenin

indicating the ingestion of *Yucca* sp. and *Agave* sp. which complements previous reports of these taxa based on dung macrobotanical fragments [155].

The range of *Nothrotheriops* extended beyond the deserts of the Southwestern United States, west to the coast at Rancho La Brea and into Northern California where it is known from three cave sites, Potter Creek, Samwell, and Hawver [156] and south to Belize [157]. Its eclectic diet permitted it to utilize multiple different habitats. Its presence in Belize may have coincided with an arid climatic interval and the presence of more xeric-adapted vegetation than exists in the area today.

The vegetation at Rancho La Brea differs significantly from other sites in the Southwestern United States with *Nothrotheriops*. Plant macrofossils from Pit 91, the only pit with *Nothrotheriops* that has been radiometrically dated, indicate that between 28–26 ka, the vegetation of the Santa Monica Plain was dominated by coastal sage scrub with pines and cypress at slightly higher elevations [158,159]. While this vegetation dates younger than the *Nothrotheriops* at 33,000 years, the preserved plant remains at least provide a general overview of the mosaic of different vegetation communities in the area. For example, a chaparral community grew on the slopes of the Santa Monica Mountains along with isolated coast redwood (*Sequoia sempervirens*) and dogwood (*Cornus californica*) in protected canyons with a riparian community that included willow (*Salix lasiolepis*), red cedar (*Juniperus* sp.) and sycamore (*Platanus racemosa*). These plant communities suggest a winter precipitation regime like the modern Mediterranean climate of coastal Southern California. However, the presence of coast redwood, now found 600 km to the north in the summer fog belt indicates a cooler, more mesic, and less seasonal climate at terminal OIS 3 than present and corroborates the pollen record. The presence of coast redwood and dogwood in Southern California and the Los Angeles Basin suggests the possibility of the existence of non-analog plant communities in the region. While the MAT at Rancho La Brea at 33,000 years is comparable to that of other sites with *Nothrotheriops* there was significantly more annual precipitation as indicated by the plants.

Stable isotope analysis of *Nothrotheriops* from two sites in Southern Nevada, Tule Springs, and Devil Peak Cave, ref. [160] is consistent with data from the dung that the sloth's diet consisted of primarily xeric vegetation. However, analysis of specimens from the Northern California cave sites (Potter Creek Cave, Hawver Cave, and Samwel Cave) indicated this species had a much more isotopically diverse diet than previously thought based on the dung samples indicating an enrichment in $^{13}$C through life, with a tendency to consume C3 or mixed vegetation as juvenile and increasing percentages of C4 vegetation as it aged. The $\delta^{13}$C for the Devil Peak individual was –4.1 and –4.7 for the Tule Springs individual while the $\delta^{18}$O was –6.9 for the former and –6.1 for the latter. In contrast, the $\delta^{13}$C values for Potter Creek Cave were –8.6 and–8.4, Samwell Cave –8.9 and–8.0, and Hawver Cave–11.4 while the $\delta^{18}$O values were–6.9 (Devil Peak), –6.1 (Tule Springs) while the values for the Northern California sites were –5.7 and –5.2 (Potter Creek Cave), –4.7 and–4.3 (Samwell Cave) and –4.2 (Hawver Cave).

The average MAT is 12 °C with a range of 6–19 °C indicates *Nothrotheriops* was the least cold tolerant of the three sloths in temperate North America and this is reflected by the lower latitudes and elevations for the majority of sites where it has been found and its range extended farther south than either *Megalonyx* or *Paramylodon* [161]. Based on an average TAP of 230 mm with a range of 124 to 897 mm calculated for multiple sites it also seemed more tolerant of drier habitat than any of the other North American sloths.

*Nothrotheriops shastensis* is known from a single locality within the Southern Mexico-Central American region, Actun Lak, Cayo District, Belize [157]. This is the southernmost record of the species and the only record of a nothrothere from Central America. Based on the detailed knowledge of its diet from dung in the Southwestern United States, its presence this far south suggests that a greater variety of habitats, including arid or xeric ones, existed in this region during the Pleistocene, and that vegetation communities underwent dramatic changes during this interval.

### 4.6.4. *Nothrotheriops* sp.—Pampas and Patagonia

A femur referred to *Nothrotheriops* is the first report of this North American genus in South America [162]. While the morphology is similar to the North American genus, the Argentinian specimen is smaller than the contemporaneous species in North America, *N. shastensis*. Based on the femur length the estimated body mass is 193 kg which is significantly less than the estimate of 492 kg for *N. shastensis*. The specimen was found in the late Pleistocene of Santa Fe Province, Argentina. The lithostratigraphic unit from which the femur was recovered is interpreted as channel bed facies generated in a low energy flow regime.

### 4.6.5. *Nothropus*—Pampas and Patagonia

Multiple nothrotheres have been reported from the late Pleistocene of Santa Fe Province, Argentina. These include *Nothropus carcaranensis*, *Nothrotherium escrivanense*, *Nothrotherium roverei*, *Nothrotherium* cf. *torresi* and a Nothrotheriinae genus and species indeterminate [163]. Given the general rarity of nothrotheres in South American late Pleistocene faunas such a high diversity seems unusual for such a small geographical area. It should be noted that most of these taxa are based on isolated specimens and often not by the same bones (e.g., *Nothropus carcaranensis* on a dentary, *Nothrotherium roverei* on a humerus, *Nothrotherium* cf. *torresi* on a partial femur), which precludes direct comparison between the specimens or an estimate of the body mass. Many of these taxa may eventually become junior synonyms if an associated skeleton is found. For a number of these specimens, there is also a lack of precise stratigraphic and geographic information which precludes any real inferences as to their paleoecology based on their depositional context.

## 5. Discussion

The composition of South American mammalian fauna as a mix of taxa originating in South America and North America was already established at the beginning of the Pleistocene (ca. 2.6 Ma), with the appearance of mammals of North American origin entering the continent via the Panama corridor during different stages of the Great American Biotic Interchange (GABI) [164,165]. At the same time, there was a reciprocal dispersal of taxa of South American origin, such as the sloths, which became established in Central and North America. The potential for dispersal in either direction resulted in an increase in biotic diversity on both continents in the Pleistocene. Changes in biodiversity were often determined by the different reactions of the species involved in climate change. On the one hand, the dispersing species often resulted in the creation of non-analog assemblages (the association of species that are today allopatric) [166,167]. The composition of the mammalian fauna dramatically changed at the end of the Pleistocene with the extinction of megamammals, including sloths, along with large carnivores, and the subsequent impact on ecosystems. The removal of a species from an ecosystem, whether due to extinction, the result of a reduction or shift in its range in response to changes in characteristics of the environment or a reduction in its population density that it can be considered functionally extinct from an ecological point of view or what [168] termed Latent Extinction can also result in the extinction of ecological interactions. Examples that could be applied to the loss of ground sloths can include predator–prey interaction, or their involvement in seed dispersal [169,170] as well as many other aspects of ecosystem function [171] related to their role as megaherbivores in an ecosystem [172].

Taxonomic closeness does not translate into similarities in ecology. The megatheriids, *Eremotherium* and *Megatherium*, are more closely related to each other than to any of the other sloths yet have very distinct ecologies reflected by their distribution as shown by [11] in their panbiogeographic track analysis of South Amerian megaherbivores. The relationship between different habitat and diets and distribution is also clearly seen in the North America sloths so the number of sloths in a fauna varies and often there is only one sloth present in a fauna [173]. The late Pleistocene distribution of *Nothrotheriops* for example is restricted to the Western United States with most records in the southwest and arid desert habitat [156]. In marked

contrast the distribution of *Eremotherium* in the United States was restricted to the east coast and Southeastern United States in subtropical habitat and extending south into Mexico into more tropical regions, so there are no sites in which these two taxa co-occur. The distribution of the other two genera, *Megalonyx* and *Paramylodon*, are much greater and extend across the United States. Both taxa may be found together in a fauna, and to different degrees may be found associated with either *Nothrotheriops* or *Eremotherium*.

The mylodontid sloths, *Glossotherium* and *Paramylodon*, are more closely related to each other than to other mylodon genera such as *Mylodon*. A comparison of the stable isotope data of these two genera as shown by $\delta^{13}$C (average –21.0‰) and an average of $\delta^{15}$N (+7.9) obtained from the North American Pleistocene *P. harlani* from Rancho La Brea [120]; the $\delta^{13}$C ($-20.5$‰) and $\delta^{15}$N (+10.5 on average) for *G. robustum* the province of Buenos Aires.

In his analysis of the late Pleistocene (Lujanian) mammal fauna of Northern Argentina, Fariña [24] examined the ecological relationship between body size, population density, and basal metabolism. Out of the 30 species in the fauna, 16 are Xenarthrans, 6 of which are sloths and are the largest taxa in the fauna except for the proboscidean, *Stegomastodon* (4311 kg), the notoungulate *Toxodon* (1187 kg), and the litoptern *Macrauchenia* (830 kg). If the ground sloths are removed from the herbivore community an estimated 4587 kg km$^2$, ca 41% of the biomass would be removed from the megaherbivoreportion of the fauna.

Changes in distribution and expansion or contraction of the range also impact population size and density. These in turn are impacted by changes in sea level during the Pleistocene and alternating exposure or flooding of the continental shelf. This tends to be primarily on the east coast of the American continents and the expansion of sloths onto the continental shelf has been documented in both North America with the recovery of *Megalonyx* [174] and *Megatherium*, *Glossotherium*, *Lestodon*, *Mylodon*, and *Catonyx* from off the coast of Brazil, Uruguay, and Argentina [175]. During the LGM the estimated maximum lowering of sea level, ca 120 m, would have increased the available land area of Buenos Aire Province, Argentina, the Rio de La Plata, and adjacent Atlantic areas of Uruguay and Southern Brazil by 60% [20]. Related to lower sea levels, the coastline of the Buenos Aires Province would have extended 150 km east of its present location ca.11,000 years B.P. [176]. At this time, regional vegetation in the area close to the modern coast would have been dominated by Pampa grassland taxa indicating that conditions were more continental than today. This increase in the availability of potential habitat would have increased the on-crop biomass for sloths as well as other species, based on the smaller area used by [24] would have increased from 11,260 kg/km$^2$ to 18,016 kg/km$^2$.

Examples of sloth taxa that would have dispersed into this newly available habitat include *Glossotherium robustum*, *Lestodon armatus*, and *Mylodon darwinii* as areas of high suitability for this taxon were identified for submerged parts of the continental shelf that were exposed during the LGM in the species distribution models [12]. The co-occurrence of these sloths at the Arroyo de Vizcaino site in Uruguay suggests the presence of vegetation indicative of open, cold to temperate habitats but with mixed patches typical of humid climates.

Species distribution models (SDMs) for the last interglacial (LIG), the LGM, and the Holocene climatic optimum (HCO) were generated for three extinct South American Pleistocene mylodontid sloths, *Glossotherium robustum*, *Lestodon armatus*, and *Mylodon darwinii* [12]. These taxa are present in multiple local faunas and their potential co-occurrence provides the basis to generate inferences on their potential distributions that can be compared with the available biome reconstructions of South America during the LGM. This permitted an analysis of their distribution patterns, ecological requirements, and inferences on their types of interaction. The results indicated these three sloths could have co-existed in the Chaco-Paraná Basin and the plains associated with the Río de la Plata. The area of overlap also included what are now submerged parts of the continental shelf that were exposed during the LGM. This would also indicate there was a drastic reduction in total available areas of preferred habitat for these three taxa with the rise in sea level at the end of the Pleistocene. Based on their co-occurrence at the Arroyo del Vizcaíno site, Uruguay,

they were utilizing vegetation indicative of open, cold to temperate habitats but with mixed patches typical of humid climates [177].

In this regard, Buenos Aires, Entre Ríos, Corrientes, Formosa, and Chaco provinces (included in the Pampean Region) are good examples of an ecotone or transitional zone between faunas of the more tropical BIR [9] to the north and Patagonian subregions to the south [164,178]. This transition zone or ecological tension zone [179] marks a transition between two distinctive biotic zones in which there is a change in the biota influenced by climatic factors and geological variation. The zone between the temperate zone of Argentina and Uruguay and the tropical zone of Brazil corresponds to node one of [11] as the area of contact between tropical and temperate taxa megafauna in the Pleistocene. This provides a basis to document the response of each species to environmental changes such as shifts in distribution and short-term intermixing of taxa that may result in non-analog faunas. Additionally, given the climatic dynamics of the late Pleistocene to Holocene transition, it would be expected that this area of contact shifted in response to climate change. However, to document this will require a good chronology of the local fauna and the individual species for this region. The species distribution model [12] showed that *Megatherium* and *Eremotherium*, have little to no overlap in their potential distributions during the LGM. Yet *Megatherium* has been reported from two localities in the state of Rio Grande do Sul, Southeastern Brazil, where it overlaps with the southern part of the range of *Eremotherium*, on the Coastal Plain [180] and both genera are present in the fauna from Caçapavo do Sul, the only site in which they cooccur [15] but these specimens have not been individually radiocarbon dated to determine how close in time they lived in the area. *Eremotherium laurillardi* is considered a key species in the BIR due to its high body weight and wide niche breadth and proposed that *E. laurillardi* was a major competitor for resources in the BIR [50]. Given the extensive range of this species, this interpretation of its ecological flexibility could be extrapolated to the species as a whole. This flexibility would allow it to occur with a greater diversity of other sloth genera than any other sloth species. Its occasional association with *M. americanum* may simply reflect its ecological flexibility, while the ecological requirements of *M. americanum* were narrower.

The upper Ribeira region in São Paulo State includes several megafauna species from both the Pampean and Intertropical provinces. In this region, the intertropical *Eremotherium laurillardi* overlaps with *Lestodon armatus* which like *Megatherium americanum* is more characteristic of the Pampas fauna to the south. The co-occurrence of these animals again supports the interpretation that the area in question is a transition zone between these two biogeographic regions, with other characteristic species of each besides the sloths overlapping. For example, *E. laurillardi* is a typical species of the intertropical zone, while *L. armatus* in contrast is a characteristic taxon of more temperate regions and the Pampas. *Lestodon* is more common in faunas from the Southern Cone South American countries but becomes rare and then absent at lower latitudes. The furthest northern occurrence of *Lestodon* is the Ribeira cave deposits and the clay deposits of Álvares Machado municipality, São Paulo State [133]. As this area is farther north, ca 24° S than the Rio Grande do Sul ca. 30.5° S the presence of a larger number of taxa typical of the intertropical region, including the sloths, *E. laurillardi*, *Nothrotherium*, *Catonyx*, and probably *Ahytherium*, suggests a greater affinity to this paleobiogeographic region. However, as noted previously, both *Nothrotherium* and *Catonyx* have been found farther south in Uruguay so changes in faunal composition with latitude tend to be gradational with species responding independently to various ecological or climatic parameters [166].

A similar transition or tension zone, although not as pronounced relative to the sloths, exists between temperate North America and tropical Southern Mexico and Central America. This transitional area occurs at the Isthmus of Tehuantepec but is not uniform as the transition from temperate to tropical is also influenced by the north–south trending mountain ranges of the Sierra Madre Occidental and Oriental which provide habitat for more northern species at lower latitudes. In this case, the two northern sloths that extend their range south of this zone inhabit two quite different habitats. *Nothrotheriops shastensis* which

extends its range south into Belize but is desert-adapted and *Eremotherium laurillardi* which is a subtropical to tropical species. Extending north of the zone is *Xibalbaonyx microcaninus* from the state of Jalisco in West–Central Mexico [181]. Given the dramatic differences in ecology and non-overlapping distribution in temperate North America, their presence in Southern Mexico and Central America occurred at separate times reflecting changes in suitable habitat for each species due to climatic changes. As *Xibalbaonyx microcaninus* is only known from one specimen, more information is needed on its ecology to determine the climatic or environmental changes that allowed this species of *Xibalbaonyx* to move northward since the other two species of this genus are found only in Yucatan.

While one genus may be widespread and its range overlaps with multiple other sloth genera, competitive exclusion can occur at the species level with various species within a genus having distinct non-overlapping ranges. A comparison of the ranges of *Catonyx cuvieri* and *C. chilense* found that while *C. cuvieri* had a wider potential distribution covering most of the continent and *C. chilensis* showed highly suitable areas in the BIR intertropical region overlapping with zones in the model, there is no locality in the fossil record where the two species co-occur [12]. The habitat preferences of these taxa indicate *C. cuvieri* is associated with tropical rainforests and *C. chilensis* to a more arid environment such as dry forests [182] or semi-desert and arid conditions [183,184]. Based on their study Varela et al. [12] propose that both species may have inhabited arid open environments but *C. cuvieri* had a greater ecological tolerance to more forested habitats.

Another example is the mutually exclusive distributions of *Glossotherium tropicorum* and *G. wegneri* in Ecuador. The former species is found exclusively in the lowlands, while the latter's distribution is at high elevations in the Andes [185,186]. However, it should be noted that some workers consider *G. wegneri* a distinct genus, *Oreomylodon*, and the two taxa may not be closely related, hence the marked differences in their habitat preference [111].

## 6. Extinction

Given the diversity of sloths in North, Central, and South America, along with those on some Caribbean islands and the variety of different habitats in which they occurred, there is certainly no doubt as to the challenge of determining a single underlying cause for the extinction of so many distinct species. While there was a period of overlap between the first appearance of humans in the ecologically distinct parts of the Americas and the subsequent disappearance of the Pleistocene megafauna in the New World, including sloths, evidence as to the degree of interaction is limited. While there is good evidence that the sloths in the Caribbean became extinct much later than those on the mainland [5] the available radiocarbon dates for each of the continental species of sloth are extremely limited, thus making correlations with climatic events or first appearance of humans in a region tenuous at best. Evidence of direct interactions between extinct sloths and humans is also extremely limited although some evidence such as butcher marks on sloth bones [187] and associated human and sloth tracks [188] do indicate that these interactions did occur. An *Eremotherium* tooth modified by humans has also been reported from Brazil [189].

While it is beyond the scope of this review to argue for an either/or scenario for the extinction of ground sloths, it does seem appropriate to try and place what is known of the ecology of sloths and their extinction within a broader context of climate and environmental change at the end of the Pleistocene. The evidence suggests that continental vertebrates responded to the abrupt temperature changes, at a millennia scale, that characterized the MIS 3 with Heinrich and Dansgaard–Oeschger events. Given the low basal metabolism and body temperature of Xenarthrans [105], such temperature changes could have had a significant impact on sloths, both physiologically and on the distribution of the forage/plant resources upon which they depended.

A general overview of deglaciation in North, Central, and South America from the LGM to the beginning of the Holocene is worth considering for the continental sloths. Given the much later extinction of the taxa on the Caribbean islands, the context of their extinction needs to be considered separately. Available paleoclimatic and paleoceanographic data

provide a framework with which to compare the pace of deglaciation and the response of glaciers to major climate events, as these same climatic events could also have more directly impacted the sloths and their habitats, the decline of glaciers may be viewed as more of a proxy rather than a driver of sloth extinction. During the Global Last Glacial Maximum (GLGM) from 26.5–19 ka the average temperatures decreased 4° to 8 °C in the Americas, but precipitation varied strongly throughout this large region. During the GLGM multiple glaciers in North and Central America achieved their maximum extent, whereas others achieved their maximum extent during the subsequent Heinrich Stadial 1 (HS-1). During the GLGM, glaciers in the Andes also expanded, but their advance was not the largest, except on Tierra del Fuego. Heinrich event 1 (17.5–14.6 ka) was a time of general glacier thickening and advancing throughout most of North and Central America. This advance also occurred in the tropical Andes, but glaciers in the temperate and subpolar Andes thinned and retreated at this time.

During the warm interval of the Bølling–Allerød interstadial (14.6–12.9 ka), glaciers retreated throughout North and Central America and, in some cases, completely disappeared. This was followed by an advancing of glaciers during the Antarctic Cold Reversal (14.6–12.9 ka) in the tropical Andes and Patagonia. During the Younger Dryas (12.9–11.7 ka) there were also small glacial advances in North America, Central America, and Northern South America (Venezuela) but glaciers in Central and Southern South America retreated during this period, except on the Altiplano where advances were driven by an increase in precipitation. There does not seem to have been any synchroneity in these events between the hemispheres, which affected not only the behavior of glaciers but also marine currents and atmospheric circulation [106].

Given the wide distribution of sloths in North, Central, and South America and their presence in a variety of habitats, it seems reasonable to assume that at various stages of these events on different continents, sloths as well as their faunal associates, would have been impacted differently and at different times. As observed by Meltzer [190], "Resolving the cause of large mammal extinctions requires greater knowledge of individual species' histories and their adaptive tolerances, a fuller understanding of how past climatic and ecological changes impacted those animals and their biotic communities, and what changes occurred at the Pleistocene–Holocene boundary that might have led to those genera going extinct at that time".

The disappearance of most large-bodied herbivores at the end of the late Quaternary would have fundamentally altered the structure and functioning of ecosystems [191]. It is reasonable to assume that different sloths had different impacts on the environments they inhabited, especially when considering the wide range in size of different taxa. The degree and type of impact could be as simple as whether a species was a browser or a grazer. Yet, we simply do not know either the type or extent of those impacts and exactly how each sloth species functioned within its respective ecosystem. The disappearance of large herbivores also fundamentally altered fire regimes and vegetation communities, and the feedback between the two [192]. High herbivore diversity and abundance can maintain heterogeneous vegetation across landscapes, whereas the decline and subsequent disappearance of large mammals may produce a homogenization of plant communities.

As there is an inverse relationship between body size and population density, the role of the smaller sloths, such as the nothrotheriids, with presumably larger populations, would have been quite different from that of the gigantic megatheriids, even though both are considered to have been browsers. As the sloths are only one component of the megaherbivores in the Pleistocene faunas of the New World, their extinction, along with that of other large herbivores would have had a cascading effect on both higher and lower trophic levels. As an important part of the prey base, the loss of herbivorous large mammals would be clearly linked to the collapse of large carnivore guilds in North, Central, and South America. Modern sloths have a distinctive invertebrate fauna associated with them [193] so it is likely that extinct sloths also had invertebrate communities with mutualistic relationships, as well as others, including the transport of seeds caught in

their hair [194] or as has been postulated in their dung [195]. As previously discussed, the disappearance of large herbivores can also severely hinder the ability of many large fruit-bearing plant species to disperse, leading to lower recruitment and in some cases near-extinction in the wild. These are just a small set of examples of the roles of the extinct sloths in ecosystems but unfortunately for which the fossil record has provided little direct evidence.

In addition to these aspects of the natural history of sloths, as well as other Pleistocene megafauna, the lack of radiocarbon dates for many of the late Pleistocene taxa should be added. Without a strong chronology, it is not possible to connect their presence or absence/extinction in a region to climatic and related environmental change or to the presence of humans and their impact on the local environment. It, therefore, does not seem reasonable to expect the extinction of sloths to have occurred during a narrow window of time, but rather more likely took place at separate times in different regions. Until there is a better chronology for the various taxa that cover the entire range of the species, it is not likely that a plausible explanation or a pattern for their extinction can be determined.

**Funding:** This research received no external funding.

**Acknowledgments:** Gerry DeIuliis, Mário Dantas, and Ascanio Rincón generously helped with providing some of the skull images of various sloth taxa. Appreciation is extended to the reviewers for their ideas and input.

**Conflicts of Interest:** The author declares no conflict of interest.

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
