# Peer review of "A Tale of Two Continents (and a Few Islands): Ecology and Distribution of Late Pleistocene Sloths"

_land, doi:10.3390/land12061192_

Round 1
Reviewer 1 Report
I find this report to be very thorough, well organized, and up to the usual high standards of the author. It is a very welcome treatment of the topic, bringing together much disparate, wide ranging information to bear on aspects of the ecology, distribution, and systematics of Pleistocene sloths. I believe there are a few places where additional citations would strengthen several of your arguments and a couple of instances where the systematics could be treated in further detail without greatly lengthening the manuscript. My suggestions in such cases are indicated as notes on the pdf.

Most of my comments on the English are directed at typos (some of which seem to have been the result of converting the document to a pdf). In a few places I have suggested changes that I feel might improve comprehension (although some a simply stylistic). All of these are indicated in the reviewed pdf.
Author Response
Reviewer 1 made the most comprehensive comments on the manuscript including a copy in which specific items were suggested and I have addressed all of these. This has included revising Table 1 as requested. modifying the captions for the figures so the genus and species are in italics. rewording certain sections which they felt were not clearly conveying the information, including the removal of nonessential text which streamlined the and the addition of new references for sections which they felt required additional documentation. I agree with their observation that there is some question as to whether the species wegneri should be in Glossotherium or Oreomylodon. However as this is not a systematics paper, the issue cannot be resolved here so I have chosen to use Oreomylodon based on the most recent study. I found the comments of Reviewer 1 very helpful and was happy to make all of the suggested modifications.
Reviewer 2 Report
I read the text with great pleasure. The article concerns research that has been skillfully set up and has clearly demarked the areas of investigation. The aims of the research are however not very clear. The background of the research is properly described and supported by an adequate amount of comparative studies.
There is a wide descriptive part on the areas of distribution of sloths and an extensive description of the fossil remains. This extensive overview is certainly the merit of the article. The research methods and problems concerning the investigation are clearly designated. Admirable is the list of literature and utilization of published research.
The researchers are adequately aware of the interpretation’s pitfalls. Personally, I think it should be more highlighted the higher vulnerability of the islands versus the mainland.
The article does not provide real solutions and explanations. Its value consists more in reviewing and appointing for the problems, as pointed out by their claim for more appropriate, or lack of, chronological dates (last sentences).
In the text, few indentations must be corrected.
Author Response
The main suggestion of reviewer 2 was to have more discussion on the higher vulnerability of island taxa to extinction. This is an important topic and there is already considerable literature. Given the scope of the topic I feel it is beyond the goal of this paper to try and address the topic as it applies to the island sloth taxa. At this time I do not feel there is sufficient data on the ecology of the island sloths to be able to fully address this issue and more data is needed in order to provide any reasonable interpretation on their extinction. As will be noted in the manuscript, the section on the Caribbean sloths is small, and was included primarily to try and be as comprehensive as possible in this review paper to at least provide some general background information in order to provide a basis for the observation that their ecology and causes of extinction must have been quite different from that of the continental sloths. It is an important topic and one that should be addressed but I do not feel it can be done adequately in this review paper. Therefore I would like to respectfully like to decline to develop this topic in any further detail. They noted the problem with the indentations for some sections (as did reviewer 1) and this has been done.
Reviewer 3 Report
In this manuscript, there is an exhaustive and interesting description of the species types of sloths, and I think follow the text could be easier with a table explicating the different characteristics: estimated body mass, food, geographic distribution, isotopic values…
The author could include a map with the distribution areas of the different sloths.
According to guide of author of this journal the references in the text should be reference numbers in square brackets [ ].
Please, be careful with the names of species. They must be in italics (pay attention on the tittles from the every paragraph)
Indicate, please, the text of every caption and the reference from they came. And if it is possible the scale close to the pictures should be useful.
I should add a paragraph of abbreviations: MAT, BIR….if you haven’t already included them in the text.

Author Response
I have changed the style of brackets as noted by the reviewer. Abbreviations are noted in the test the first time at term is used for which an abbreviation is used later. As the skull illustrations are for illustrative purposes and not illustrate any specific differences of taxa as in a systematic paper I do not feel that a scale is a critical part of the paper.
Reviewer 4 Report
This manuscript represents an important contribution regarding the current knowledge about Late Pleistocene sloths in the Americas. The author provides an extensive review of all the genera and discuss the different species that have been proposed. Furthermore, the revision covers many ecological and biogeographical aspects, which are often not discussed by many authors in favor to systematic revisions.
I think that the manuscript needs some editing in other to improve readability, as there are many small errors and typos, as well as formatting issues.
Furthermore, I think that an extra figure showing the geographic distribution of the taxa and the region's discussed in the text would greatly improve the presentation of the information for the readers.
As mentioned before, there are many small errors and typos in the text. There are also some formatting issues with paragraphs, and some issues with taxa names not in italics or misspelled.
Please check throughfully.
Author Response
Given the number of taxa covered in this review, the inclusion of maps for each species would add to the length. However, if the editors feel that the inclusion of maps as suggested by the reviewer would be acceptable I would be happy to includes them.
I have made the edits to the English as per suggestions of Reviewer 1.